# CBMA: Improving Conformal Prediction through Bayesian Model Averaging

**Pankaj Bhagwat**
Department of Mathematical
and Statistical Sciences,
University of Alberta, AB, Canada
pbhagwat@ualberta.ca

**Linglong Kong**
Department of Mathematical
and Statistical Sciences,
University of Alberta, AB, Canada
lkong@ualberta.ca

**Bai Jiang**
Department of Mathematical
and Statistical Sciences,
University of Alberta, AB, Canada
bei1@ualberta.ca

## Abstract

Conformal prediction has emerged as a popular technique for facilitating valid predictive inference across a spectrum of machine learning models, under minimal assumption of exchangeability. Recently, Hoff (2023) showed that full conformal Bayes provides the most efficient prediction sets (smallest by expected volume) among all prediction sets that are valid at the $(1 - \alpha)$ level if the model is correctly specified. However, a critical issue arises when the Bayesian model itself may be mis-specified, resulting in prediction set that might be suboptimal, even though it still enjoys the frequentist coverage guarantee. To address this limitation, we propose an innovative solution that combines Bayesian model averaging (BMA) with conformal prediction. This hybrid not only leverages the strengths of Bayesian conformal prediction but also introduces a layer of robustness through model averaging. Theoretically, we prove that the resulting prediction set will converge to the optimal level of efficiency, if the true model is included among the candidate models. This assurance of optimality, even under potential model uncertainty, provides a significant improvement over existing methods, ensuring more reliable and precise uncertainty quantification.

## 1 Introduction

Conformal prediction is a distribution-free uncertainty quantification method that generates prediction sets with valid coverage under minimal assumptions of exchangeability (Vovk et al., 2005). By building on existing machine learning techniques, conformal prediction uses available data to establish valid prediction set $C_\alpha(X_{n+1})$ for new instances $X_{n+1}$ with coverage guarantee $P(Y_{n+1} \in C_\alpha(X_{n+1})) \geq 1 - \alpha$. Prediction sets that include the ground truth with high probability are essential for high-stakes applications, such as autonomous vehicles (Bojarski et al., 2016) and clinical settings (Esteva et al., 2017). However, it is also preferable for the prediction sets $C_\alpha(X_{n+1})$ to be efficient i.e. as small as possible, as smaller sets are more informative. Recently, Hoff (2023) investigated the optimal efficiency of full conformal Bayes prediction sets under a correctly specified Bayesian model, showing that they yield prediction sets with the minimum expected volume at a given target coverage level $(1 - \alpha)$. In addition, Fong & Holmes (2021) introduced scalable methods for generating full conformal Bayes prediction sets applicable to any Bayesian model, highlighting the advantages of conformal Bayesian predictions over traditional Bayesian posterior predictive sets, especially in the presence of model misspecification.

However, constructing or selecting the correct Bayesian model to obtain optimally efficient conformal prediction sets is a challenging task. Although conformal prediction sets provide finite-sample coverage guarantees for any model, they may fail to achieve volume optimality when the underlying

model is misspecified. This challenge underscores a broader limitation of traditional conformal prediction methods, which typically rely on a fixed machine learning model to generate prediction sets with a predetermined marginal coverage level. Given the vast array of possible predictive models for a given problem, and the fact that conformal methods yield valid sets for all of them, identifying the most appropriate model becomes inherently difficult. This issue of model uncertainty has long been a persistent and underexplored problem in the conformal prediction literature.

**Our Contributions:** In this paper, we provide an innovative solution in the form of Conformal Bayesian model averaging (CBMA) to the challenging issue of constructing efficient conformal prediction sets when there is model uncertainty in the Bayesian framework. Our proposed CBMA prediction method seamlessly combines conformity scores of each Bayesian model in order to construct a single conformal prediction set. This paves the way for incorporating model averaging procedures into conformal prediction framework which is currently lacking in the literature. To the best of our knowledge, our CBMA approach is the first method which combines conformity scores from diverse models to construct valid conformal prediction sets and requires no data splitting (data efficient). Our choice of conformity scores in the form of posterior predictive densities is also optimal and leads to the most efficient prediction set when the underlying Bayesian model is true (Hoff, 2023). Our CBMA method can construct full conformal prediction sets given samples for model parameters of each model from their posterior distributions and posterior model probabilities, which can be obtained as the output of BMA. Theoretically, we also prove the optimal efficiency achieved by conformal prediction sets constructed using CBMA method as the sample size increases. Such guarantee of optimal efficiency even under model uncertainty provides an improvement over existing methods. Finally, our method incorporates both data and model uncertainty into the construction of prediction sets which enhances the reliability of the predictions.

## 2 PRELIMINARIES

We consider a collection of i.i.d. observations $Z_{1:n} = \{(X_i, Y_i)\}_{i=1}^n$, where each observation $(X_i, Y_i) \in \mathbb{R}^d \times \mathbb{R}$ is a covariate-response pair. We begin by summarizing full conformal prediction framework (Vovk et al., 2005), conformal Bayes (Fong & Holmes, 2021) and Bayesian model averaging (BMA) (Raftery et al., 1997).

### 2.1 FULL CONFORMAL PREDICTION

In the full conformal prediction method, candidate labels $y_{n+1}$ are included in $C_\alpha(X_{n+1})$ if the resulting pair $(X_{n+1}, y_{n+1})$ is sufficiently similar to the examples in $Z_{1:n}$. A conformity score $\sigma_i$

$$\sigma_i := \sigma(Z_{1:n+1}; Z_i),$$

takes as input a set of data points $Z_{1:n+1} = Z_{1:n} \bigcup Z_{n+1}$ where $Z_{n+1} = \{(X_{n+1}, y_{n+1})$ , and computes how similar the data point $Z_i$ is for $i = 1, \ldots, n+1$. The key property of any conformity score is that it must be exchangeable in the first argument, meaning the conformity core for $Z_i$ remains unchanged under permutation of $Z_{1:n+1}$. We refer a conformity score with this property as a valid conformity score. Assuming $Z_{1:n+1}$ is exchangeable, $\sigma_{1:n+1}$ will also be exchangeable, and its rank will be uniformly distributed among $\{1, \ldots, n+1\}$ (assuming continuous $\sigma_{1:n+1}$). This implies that the rank of $\sigma_{n+1}$ is a valid p-value. For a given value $Y_{n+1} = y$ (with $X_{n+1}$ known), we can denote the rank of $\sigma_{n+1}$ among $\sigma_{1:n+1}$ as

$$r(y) = \frac{1}{n+1} \sum_{i=1}^{n+1} \mathbf{1}(\sigma_i \leq \sigma_{n+1}).$$

For miscoverage level $\alpha$, the full conformal predictive set, $C_\alpha(X_{n+1}) = \{y \in \mathcal{Y} : r(y) > \alpha\}$, ensures the desired frequentist coverage $\mathbf{P}(Y_{n+1} \in C_\alpha(X_{n+1})) \geq 1 - \alpha$, where $\mathbf{P}$ is over $Z_{1:n+1}$. A formal proof can be found in Vovk et al. (2005), Chapter 8.7 . For continuous $\sigma_{1:n+1}$, Lei et al. (2018), Theorem 1 show that the conformal predictive set does not significantly over-cover.

### 2.2 FULL CONFORMAL BAYES

In the Bayesian prediction, given the model likelihood $p_\theta(y|x)$ and prior on parameters, $\pi(\theta)$ for $\theta \in \mathbb{R}^p$, the Bayesian posterior predictive distribution for the response at new instance

$X_{n+1} = x_{n+1}$ conditioned on the observed data $Z_{1:n}$ represents subjective and coherent beliefs. As noted by Draper (1995) and Fraser (2011), Bayesian predictive sets such as Highest posterior density predictive credible sets are poorly calibrated in the frequentist sense. This has consequences for the robustness of such approaches and trust in using Bayesian models to aid decisions. Conformal inference for calibrating Bayesian models was previously suggested in Vovk et al. (2005); Wasserman (2011). In a Bayesian model, it is natural to consider the posterior predictive density as the conformity score:

$$\sigma_i = p(Y_i|X_i, Z_{1:n+1}) = \int p_\theta(y_i|x_i)\pi(\theta|Z_{1:n+1}) \, \mathrm{d}\theta$$

This is a valid conformity score, $\sigma_i$ is indeed invariant to the permutation of $Z_{1:n+1}$ due to the form of $\pi(\theta|Z_{1:n+1}) \propto \pi(\theta) \prod_{i=1}^{n+1} p_\theta(y_i|x_i)$. This method is referred to as Conformal Bayes in Fong & Holmes (2021).

## 2.3 BMA: BAYESIAN MODEL AVERAGING

In detail discussion on BMA can be found in the seminal work of Hoeting et al. (1999), Wasserman (2000), Raftery et al. (1997). Let $X \in \mathbb{R}^d$ denote the vector consisting of all predictors under consideration and $Y \in \mathbb{R}$ denote the response variable. In the supervised learning tasks, our goal is to construct a predictive model for $Y$ based on $X$ via some model $\mathcal{M}$. The Bayesian model $\mathcal{M}$ in such scenario can be represented as

$$Y|X, \theta, \mathcal{M} \sim p_\theta(y|x) \; ; \; \theta|\mathcal{M} \sim \pi(\theta) \; ,$$

where $\theta$ denotes the parameters of the model $\mathcal{M}$, $\pi(\theta)$ denotes the prior on the parameters and $p_\theta(y|x)$ denotes the likelihood function under the model $\mathcal{M}$.

Now suppose, we have several competitive models $\mathcal{M}_k$ with parameters $\theta_k \in \Theta_k$ leading to model space $\boldsymbol{\mathcal{M}} = \{\mathcal{M}_1, \ldots, \mathcal{M}_K\}$. The Bayesian model for each model is described as:

$$Y|X, \theta_k, \mathcal{M}_k \sim p_{\theta_k}(y|x) \; ; \quad \theta_k|\mathcal{M}_k \sim \pi_k(\theta_k).$$

If $\Delta$ is a quantity of interest, say predicted values at a covariate $x \in \mathbb{R}^d$, then the expected value of $\Delta$ given the data $Z_{1:n}$ is obtained by first finding the posterior expectation of $\Delta$ under each model, and then weighting each expectation by the posterior probability of the model in the BMA framework:

$$\mathbf{E}[\Delta|Z_{1:n}] = \sum_{k=1}^{K} p(\mathcal{M}_k|Z_{1:n})\mathbf{E}[\Delta|Z_{1:n}, \mathcal{M}_k].$$

Similarly, the posterior distribution for $\Delta$ can be represented as a mixture distribution over all models,

$$p(\Delta|Z_{1:n}) = \sum_{k=1}^{K} p(\mathcal{M}_k|Z_{1:n})p(\Delta|Z_{1:n}, \mathcal{M}_k) \tag{1}$$

where $p(\Delta|Z_{1:n}, \mathcal{M}_k)$ is the posterior distribution of $\Delta$ under model $\mathcal{M}_k$. In equation 1 the posterior probability of model $\mathcal{M}_k$ is given by

$$p(\mathcal{M}_k|Z_{1:n}) = \frac{m(Z_{1:n}|\mathcal{M}_k)p(\mathcal{M}_k)}{\sum\limits_{k=1}^{K} m(Z_{1:n}|\mathcal{M}_k)p(\mathcal{M}_k)},$$

where $p(\mathcal{M}_k)$ is the prior probability that $\mathcal{M}_k$ is the true model and $m(Z_{1:n}|\mathcal{M}_k)$ is the marginal likelihood under model $\mathcal{M}_k$,

$$m(Z_{1:n}|\mathcal{M}_k) \quad = \int \prod_{i=1}^{n} p_{\theta_k}(Y_i|X_i)\pi_k(\theta_k) \, d\theta_k$$

obtained by integrating over the prior distributions for model specific parameters. Note that all conditional expressions in this section denotes condition on $Z_{1:n} = \{(X_i, Y_i)\}_{i=1}^{n}$.

## 2.4 PROBLEM SETUP

Our objective is to construct a conformal prediction set $C_\alpha(X_{n+1}) \subseteq \mathbb{R}$ that provides a range of possible responses for a test point $X_{n+1}$ based on a collection of i.i.d. observations $Z_{1:n} = \{(X_i, Y_i)\}_{i=1}^n$. We focus on Bayesian prediction based on the training data $Z_{1:n}$. Suppose we have $K$ competing models for this data, denoted as $\mathcal{M}_1, \ldots, \mathcal{M}_K$, forming a model space $\boldsymbol{\mathcal{M}} = \{\mathcal{M}_1, \ldots, \mathcal{M}_K\}$. For each model $\mathcal{M}_k$ with $k = 1, 2, \ldots, K$, the data likelihood is specified as $p_{\theta_k}(y_i \mid x_i)$, where $\theta_k$ represents the parameters of model $\mathcal{M}_k$. Let $\pi_k(\theta_k)$ denote the prior distribution on the parameters $\theta_k$ of model $\mathcal{M}_k$.

The key question is: with multiple candidate models in $\boldsymbol{\mathcal{M}}$, how can we combine their predictions to create a single conformal prediction set? Additionally, we aim to construct prediction sets using a transductive conformal approach, rather than relying on data-splitting inductive methods, which are often less efficient in terms of information usage. We propose to average conformity scores from each model to construct combined conformity score. This task presents several challenges including the issue of the same dataset being utilized to train individual models, the conformity scores are not independent. Also, note that, we need to combine conformity scores in a fashion that assigns higher weights to the conformity scores of the true model, if it is included in the model space. Thus, weights used for averaging should be data dependent.

## 2.5 RELATED WORKS

Conformal Bayes has been a topic of research interest for a long time. But some recent advances in terms of optimal efficiency of such prediction sets and scalable methods for the construction of full conformal sets are noteworthy. Hoff (2023) demonstrated that the log predictive densities are the optimal conformity measure to construct conformal prediction sets (minimum expected volume) when the underlying Bayesian model is true. Fong & Holmes (2021) recently provided a scalable Full conformal Bayes algorithm to construct a Bayes conformal set efficiently. They also explored the benefits of conformal Bayes prediction over Bayesian predictive sets when the model is misspecified. But, the conformal prediction set they construct may be inefficient. We address this issue by incorporating model averaging technique of Bayesian model averaging (BMA) into the conformal framework. BMA has been proposed as a natural Bayesian solution to account for model uncertainty (Hoeting et al., 1999; Wasserman, 2000; Raftery et al., 1997). Some recent works proposing model aggregation strategies to combine conformal prediction sets or p-values include Gasparin & Ramdas (2024a), Gasparin & Ramdas (2024b), and Yang & Kuchibhotla (2024). Gasparin & Ramdas (2024b) proposed conformal set aggregation by majority vote strategy, and proved that a merged set suffers a loss in coverage i.e. merged conformal set will have at least $1 - 2\alpha$ marginal coverage if the individual sets have $1 - \alpha$ marginal coverage. Furthermore, size of such merged sets is shown be no larger than the maximum size among the given sets. However, they do not show if the merged set will achieve optimal efficiency under certain conditions. Yang & Kuchibhotla (2024) provide efficiency first conformal prediction method but this approach suffers from a coverage loss in the aggregation step. Many existing works are available for aggregating inductive conformal predictors which require data splitting (Vovk, 2015; Linusson et al., 2017; Carlsson et al., 2014; Toccaceli & Gammerman, 2019) and are based on combining p-values which may not be well-calibrated. Linusson et al. (2020) proposed the use of out-of-bag ensemble conformal predictors. But, their theoretical validity of coverage is only approximate and evaluated solely based on empirical studies. Few methods attempt to merge conformal scores from various models under the assumption that distinct data are employed to train each model (Gauraha & Spjuth, 2021), thereby constructing conformal sets accordingly. Overall, common limitations of existing methods include all or some of the following: the requirement data splitting or access to hold-out set for calibration, which may not be feasible for small sample sizes, the lack of coverage in the aggregation step, not targeted to achieve efficient prediction sets (instead focus on efficient set aggregation which may not lead to narrower prediction sets), and no theoretical guarantees of coverage and efficiency. To overcome these pitfalls of the existing approaches, we propose CBMA framework which (1) provide theoretical guarantee of efficiency of conformal prediction sets under model uncertainty, (2) allows the use of full data for all model fits and computing conformity scores for each model as well as aggregation weights are data and prior dependent, which are computed using the same data. Thus, there is no efficiency loss by data splitting at any step, and (3) no coverage loss in the aggregation process.

## 3 CBMA: CONFORMAL BAYESIAN MODEL AVERAGING

In this section, we describe our CBMA approach to construct conformal predictive sets. For each model $\mathcal{M}_k$ in the model space $\mathcal{M} = \{\mathcal{M}_1, \ldots, \mathcal{M}_K\}$, we consider Bayesian prediction using training data $Z_{1:n}$ for an outcome of interest $Y_i \in \mathbb{R}$ and covariates $X_i \in \mathbb{R}^d$. Given a model likelihood $p_{\theta_k}(y_i|x_i)$ and prior on parameters, $\pi_k(\theta_k)$ for $\theta_k \in \Theta_k$, the posterior predictive distribution for the response at a new $X_{n+1} = x_{n+1}$ takes on the form

$$p_{\mathcal{M}_k}(y|x_{n+1}, Z_{1:n}) = \int p_{\theta_k}(y|x_{n+1})\pi_k(\theta_k|Z_{1:n}) \, \mathrm{d}\theta_k,$$

where $\pi_k(\theta_k|Z_{1:n})$ is the Bayesian posterior and $\pi_k(\theta_k|Z_{1:n}) \propto \pi_k(\theta_k) \prod_{i=1}^{n} p_{\theta_k}(y_i|x_i)$. We first fit $K$ candidate models using BMA. Following Fong & Holmes (2021), full conformal Bayes prediction sets are constructed using posterior predictive densities as conformity measures:

$$\sigma_i^{\mathcal{M}_k} = p_{\mathcal{M}_k}(Y_i|X_i, Z_{1:n+1})$$

This is a valid conformity score, $\sigma_i^{\mathcal{M}_k}$ is indeed invariant to the permutation of $Z_{1:n+1}$ due to the form of $\pi(\theta|Z_{1:n})$. For each model $\mathcal{M}_k$, we employ the Add-One-In importance sampling strategy as described by Fong & Holmes (2021) once we obtain posterior samples $\theta_k^{1:T} \sim \pi_k(\theta_k|Z_{1:n})$ for a large $T$ using MCMC. Specifically, for $Y_{n+1} = y$ and $\theta_k^{1:T} \sim \pi_k(\theta_k|Z_{1:n})$, we can compute:

$$\hat{\sigma}_i^{\mathcal{M}_k} = \hat{p}_{\mathcal{M}_k}(Y_i|X_i, Z_{1:n+1}) = \sum_{t=1}^{T} \tilde{w}_k^{(t)} p_{\theta_k^{(t)}}(Y_i|X_i), \tag{2}$$

where weights $\tilde{w}_k^{(t)}$ are of the form

$$w_k^{(t)} = p_{\theta_k^{(t)}}(y|x_{n+1}), \quad \tilde{w}_k^{(t)} = \frac{w_k^{(t)}}{\sum_{t=1}^{T} w_k^{(t)}}.$$

We can also compute posterior model probabilities $p(\mathcal{M}_k|Z_{1:n})$ based on the observations $Z_{1:n}$ so that we don't need to refit the models.

Finally, we aggregate individual model conformity scores $\sigma_i^{\mathcal{M}_k}$ to construct a weighted combination $\sigma_i^{CBMA}$ defined as:

$$\sigma_i^{CBMA} = \sum_{k=1}^{K} q_k(Z_{1:n}, Z_{n+1})\sigma_i^{\mathcal{M}_k}, \tag{3}$$

where

$$q_k(Z_{1:n}, Z_{n+1}) = \frac{p(\mathcal{M}_k|Z_{1:n})p_{\mathcal{M}_k}(y|x_{n+1}, Z_{1:n})}{\sum_{k=1}^{K} p(\mathcal{M}_k|Z_{1:n})p_{\mathcal{M}_k}(y|x_{n+1}, Z_{1:n})}. \tag{4}$$

The aggregation weights are proportional to the posterior probabilities of the models and predictive posterior density of new test instance $(y, x_{n+1})$ conditioned on the observed data $Z_{1:n}$. Note that, we can easily obtain both the terms using posterior samples for parameters based on $Z_{1:n}$ and BMA also provides posterior model probabilities in the output. Now combined score $\sigma_i^{CBMA}$ can be computed as

$$\hat{\sigma}_i^{CBMA} = \frac{\sum_{k=1}^{K} \hat{p}(\mathcal{M}_k|Z_{1:n})\left(\frac{1}{T}\sum_{t=1}^{T} w_k^{(t)}\right)\hat{\sigma}_i^{\mathcal{M}_k}}{\sum_{k=1}^{K} \hat{p}(\mathcal{M}_k|Z_{1:n})\frac{1}{T}\sum_{t=1}^{T} w_k^{(t)}}. \tag{5}$$

Note that because of the data dependent weights $q(Z_{1:n}, Z_{n+1})$ in the construction of $\sigma_i^{CBMA}$, it is not clear if it is a valid conformity score in order to construct conformal prediction set. In Lemma 1, we show that $\sigma_i^{CBMA}$ is indeed a conformity measure satisfying the exchangeability criterion. Then, we construct full conformal prediction sets with coverage guarantee. We call this method as Conformal Bayesian model averaging (CBMA). We summarize our method in Algorithm 1.

---

**Algorithm 1** CBMA: Conformal Bayesian Model averaging

---

**Input:** Observed data is $Z_{1:n}$; test point $X_{n+1}$; Specify miscoverage level $\alpha$ ; for every model $\mathcal{M}_k \in \mathcal{M}$, Model likelihood $p_{\theta_k}(y|x)$ and prior $\pi_k(\theta_k)$.
**Output:** Return $C_\alpha^{BMA}(X_{n+1})$, Individual sets $C_\alpha^{M_k}(X_{n+1})$

(Run the usual BMA)
**for** $k \in 1 : K$ **do**
    Obtain posterior samples $\theta_k^{1:T} \sim \pi_k(\theta_k|Z_{1:n})$ through MCMC
    Compute $\hat{p}(\mathcal{M}_k|Z_{1:n})$
**end for**
**for** $y \in Y_{Grid}$ **do**
    Compute $\sigma_{1:n}^{M_k}$ and $\sigma_{n+1}^{M_k}$ using (2).
    Store the rank, $r_k(y)$ , of $\sigma_{n+1}^{M_k}$ among $\sigma_{1:n+1}^{M_k}$.
    Compute $\sigma_{1:n}^{CBMA}$ and $\sigma_{n+1}^{CBMA}$, using (3) and (4).
    Store the rank, $r_{BMA}(y)$ , of $\sigma_{n+1}^{CBMA}$ among $\sigma_{1:n}^{CBMA}$.
**end for**
**Return:** Set $C_\alpha^{CBMA}(X_{n+1}) = \{y \in Y_{Grid} : r_{CBMA}(y) > \alpha\}$; Individual Sets $C_\alpha^{M_k}(X_{n+1}) = \{y \in Y_{Grid} : r_k(y) > \alpha\}$

---

## 4 THEORETICAL PROPERTIES

In the Lemma 1, we show that aggregated score $\sigma_i^{CBMA}$ in (7) is indeed a valid conformity score. Our proof of Lemma 1 relies on an alternative representation for $\sigma_i^{CBMA}$ which can be interpreted as the posterior predictive density under the hierarchical Bayesian model:

$$Y|X, \theta_k, \mathcal{M}_k \sim p_{\theta_k}(y|x) \quad \theta_k|\mathcal{M}_k \sim \pi_k(\theta_k) ; \quad \mathcal{M}_k \sim p(\mathcal{M}_k). \tag{6}$$

**Lemma 1** *The aggregated score $\sigma_i^{CBMA}$ can be rewritten as*

$$\sigma_i^{CBMA} = \sum_{k=1}^K p(\mathcal{M}_k|Z_{1:n+1}) p_{\mathcal{M}_k}(Y_i|X_i, Z_{1:n+1}) = \sum_{k=1}^K p(\mathcal{M}_k|Z_{1:n+1}) \sigma_i^{\mathcal{M}_k}. \tag{7}$$

*This is the posterior predictive density under the hierarchical model (6) and hence, is a valid conformity measure. Here $p(\mathcal{M}_k|Z_{1:n+1})$ is the posterior model probability of $\mathcal{M}_k$ conditional on $Z_{1:n+1}$.*

**Proof:** We provide the details of obtaining alternative representation in (7) for aggregated scores given in (3) in Appendix A.1. Hierarchical model (6) is fully Bayesian with two kinds of parameters: the parameters $\theta_k$ of each of the model $\mathcal{M}_k$ and the models $\mathcal{M}_k$'s themselves. Thus, we note that for i.i.d. observations $Z_{1:n+1} = \{(Y_i, X_i)\}_{i=1}^{n+1}$ from the hierarchical model (6) are exchangeable. Also, the conformity measure considered is the posterior predictive density function, which is invariant to the permutation of $Z_{1:n+1}$. This follows by observing that the likelihood under each model is $\prod_{i=1}^{n+1} p_{\theta_k}(Y_i|X_i)$ is invariant under the permutations of $Z_{1:n+1}$. Thus, this is a valid conformity measure.

Once we have established $\sigma_i^{CBMA}$ as a valid conformity measure, we can define the conformal prediction set as

$$r_{CBMA}(y) = \frac{\sum_{i=1}^{n+1} \mathbf{1}(\sigma_i^{CBMA} \leq \sigma_{n+1}^{CBMA})}{n+1}, \quad C_\alpha^{CBMA}(X_{n+1}) = \{y \in \mathcal{Y} : r_{CBMA}(y) > \alpha\}. \tag{8}$$

**Theorem 1** *Assume that $Z_{1:n+1} = \{(Y_i, X_i)\}_{i=1}^{n+1}$ are exchangeable, and the conformity measure $\sigma_i^{CBMA}$ as in (7) is invariant to the permutation of $Z_{1:n+1}$. We then have*

$$\mathbf{P}\left(Y_{n+1} \in C_\alpha^{CBMA}(X_{n+1})\right) \geq 1 - \alpha,$$

where, $C_\alpha(X_{n+1})$ is defined in (8) and $\mathbf{P}$ is over $Z_{1:n+1}$. Moreover, if the conformity measures $\{\sigma_i^{CBMA}\}_{i=1}^n$ are almost surely distinct, then we have

$$\mathbf{P}\left(Y_{n+1} \in C_\alpha^{CBMA}(X_{n+1})\right) \leq 1 - \alpha + \frac{1}{n+1}.$$

Our next result demonstrates the convergence of CBMA prediction sets to the conformal prediction sets based on the true model as the sample size increases, if the true model is included in the model space $\mathcal{M}$. The intuition behind this result is grounded in the established observation that BMA approaches the true model as the sample size $n$ becomes large, provided the true model is within the considered model space (Le & Clarke, 2022). This convergence ensures that the CBMA prediction sets become increasingly accurate and reliable, mirroring the properties of conformal prediction sets derived from the true model. The detailed proof of this result is provided in the Appendix A.2.

**Theorem 2** *Suppose true model $\mathcal{M}_{true}$ is in the model space $\mathcal{M} = \{\mathcal{M}_i\}_{i=1}^k$. Under the hypothesis of Theorem 3 in Le & Clarke (2022), as $n \to \infty$, under $\mathcal{M}_{true}$, we have $q_k(Z_{1:n}, Z_{n+1}) \to 1$ in probability when $\mathcal{M}_k = \mathcal{M}_{true}$ and $q_k(Z_{1:n}, Z_{n+1}) \to 0$ in probability when $\mathcal{M}_k \neq \mathcal{M}_{true}$. Thus, for any $i$, the conformity score $\sigma_i^{CBMA}$ converges in probability to conformity score $\sigma_i^{M_{true}}$ under true model. Furthermore, let $V_{CBMA}(X_{new}, Z_{1:n})$ denote the volume of the CBMA prediction set and $V_{true}(X_{new}, Z_{1:n})$ denote the volume of the optimal conformal prediction set under the true model $\mathcal{M}_{true}$. Then, $\lim_{n\to\infty} |\mathbb{E}V_{CBMA}(X_{new}, Z_{1:n}) - \mathbb{E}V_{true}(X_{new}, Z_{1:n})| = 0$.*

**Remark 1** *Theorem 2 implies that, as the sample size $n$ becomes large, our CBMA prediction set will be similar to conformal Bayes prediction set based on true model. Thus, the expected volume of the CBMA prediction set converges to the expected volume of the conformal Bayes prediction set based on the true model. It is important to note that the expected volume of the conformal prediction set under the true model is optimal in terms of minimum expected volume (Hoff, 2023). Therefore, as $n$ grows, our CBMA prediction set becomes as efficient as the optimal conformal prediction set derived from the true model. This convergence ensures that the CBMA prediction set not only aligns with the true model but also maintains efficiency, making it a powerful tool for practical applications where model uncertainty is present.*

**Remark 2** *In the case of small sample sizes, inferring the true model for the data generation process is challenging and often impossible. In such scenarios, our method can enhance the performance of predictions by leveraging the improved predictive power of BMA. Note that BMA has been shown to outperform individual models in terms of log scores (Raftery et al., 1997).*

**Remark 3** *When the true model is not in the model space, we note that $q_k(Z_{1:n}, Z_{n+1}) \to 1$ in probability as $n \to \infty$ for $\mathcal{M}_{k^*}$ such that $D_{KL}(p_{\mathcal{M}_{true}}||p_{\mathcal{M}_{k^*}}) = \inf_{k=1,\ldots,K} D_{KL}(p_{\mathcal{M}_{true}}||p_{\mathcal{M}_k})$, where $D_{KL}(\cdot||\cdot)$ denotes the Kullback-Leibler (KL) divergence. This follows from general posterior concentration results (White, 1982; Berk, 1966; Ramamoorthi et al., 2015). Thus, if model space is large enough to contain a model in the KL divergence neighborhood of the true model, our CBMA approach will construct near-optimal conformal prediction sets.*

## 5 SIMULATION STUDY

We evaluate the performance of our conformal Bayesian model averaging prediction method in two experimental settings: $(i)$ when true model is a part of the model space, and $(ii)$ when true model is not included in the model space. In the former case, our numerical results show that the performance of our methodology is similar to that of conformal Bayes and Bayes predictive sets based on true model. This is in accordance with the results obtained by Fong & Holmes (2021) and Hoff (2023) but our method also incorporates model uncertainty adding an extra layer of robustness to the predictive inference and elevate reliability. In the second scenario, we note that our methodology leads to construction of valid prediction sets with smaller average size as compared to conformal Bayes prediction sets and Bayesian sets for individual models. This demonstrates the advantage offered by our proposed method over conformal Bayes by Fong & Holmes (2021). The details of our simulation experiments are given in the Appendix A. All codes and results are provided in the supplementary material.

## 5.1 QUADRATIC MODEL

We draw independent and identically distributed (i.i.d.) samples $x_i \sim Unif(0,1)$, for $i = 1, \ldots, n$. The response $y$ is then generated through following model:

$$y_i = \beta_0 + \beta_1 x_i + \beta_2 x_i^2 + \epsilon_i ;$$

$$\beta_0 \sim \mathcal{N}(0, 0.25) , \ \beta_1 \sim \mathcal{N}(1, 0.25) , \ \beta_2 \sim \mathcal{N}(0.5, 0.25) , \ \epsilon \sim \mathcal{N}(0, 0.2).$$

We consider three models: Model 1 ($\mathcal{M}_1$) represents the true model, Model 2 ($\mathcal{M}_2 : y_i = \beta_0 + \beta_1 x_i + \epsilon_i$) is a linear model derived by omitting the quadratic term from Model 1, and Model 3 ($\mathcal{M}_3 : y_i = \beta_0 + \beta_2 x_i^2 + \epsilon_i$) includes only an intercept and a quadratic term. In this experiment, we set the miscoverage level $\alpha = 0.20$. We use 40% of the total sample size $n$ as the test set with the remaining 60% forming the training set. The summary of the results of the different prediction sets is given in Table 1 (for sample sizes $n = 100$ and $n = 200$), provided in A.4.1.

We observe that, all conformal prediction sets (which are intervals here) achieves the target coverage in all cases. Also, as observed by Fong & Holmes (2021), conformal Bayes prediction sets have shorter lengths as compared to their corresponding Bayesian prediction sets, for both true and misspecified models. Our proposed CBMA method provides prediction sets which have similar average lengths as that of conformal Bayes based on true model. Note that CBMA also incorporates model uncertainty as opposed to conformal Bayes, hence, leads to reliable predictive inference and provides more precise uncertainty quantification. Furthermore, as shown in Figure 1, CBMA achieves optimal mean length, equal to CB under model $\mathcal{M}_1$, even at smaller sample sizes $n = 100, 200$, demonstrating practical utility of our CBMA approach.

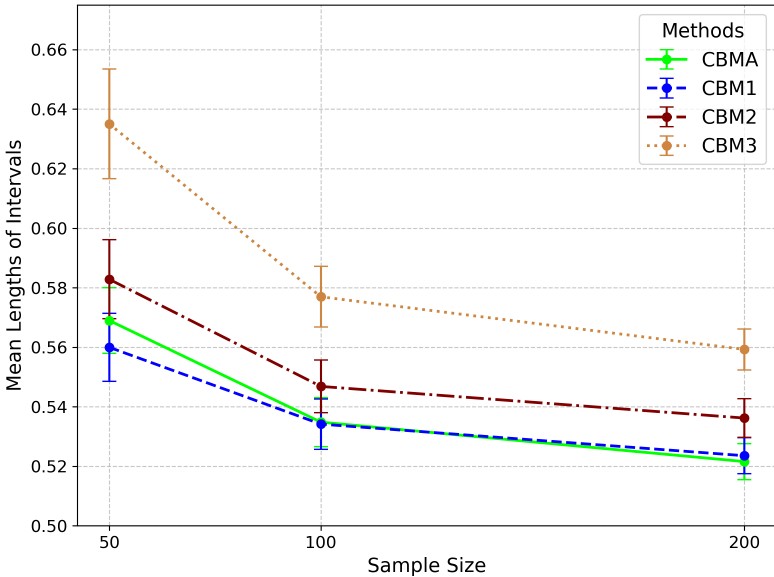

Figure 1: Quadratic model: Comparison of CBMA and CBM1 for different sample sizes.

## 5.2 APPROXIMATION USING HERMITE POLYNOMIALS

In this section, we consider a data generating mechanisms similar to Lu & Su (2015). Let $X_i \sim$ Weibull$(1, 1)$ are i.i.d. The data is generated from the following model

$$Y_i = \theta \frac{\exp\{X_i\}}{1 + \exp\{X_i\}} + \epsilon_i.$$

Here, we consider $\epsilon_i = (0.01 + X_i)\zeta_i$ and $\zeta_i \sim \mathcal{N}(0, 1)$. This is a scenario of heteroskedasticity. In this study, we employ Hermite polynomials to approximate the unknown function $\theta \frac{\exp\{X_i\}}{1+\exp\{X_i\}}$. We build a model space consisting of $K$ models with the covariates $X_{ij}, j = 1, 2, \ldots$,

$$X_{ij} = (X_i - \bar{X})^{j-1} \exp\left\{ -\frac{(X_i - \bar{X})^2}{2s_X^2} \right\}, j = 1, 2, \ldots,$$

where $\bar{X}$ and $s_X^2$ represent the sample mean and standard deviation of the set $X_i$, respectively. In our simulation experiment, we set $1 - \alpha = 0.8$. We use $40\%$ of the total sample size as the test set, with the remaining $60\%$ forming the training set ($n_{train} = 0.60 \times n$). The number of models considered is $K = 11$. We consider nested model sequence, i.e. for $k = 1, \ldots, K$, model $\mathcal{M}_k$ includes covariates $X_{i1}, \ldots, X_{ik}$. We summarize our results based on $E = 50$ experiments for samples size $n = 100$ in Figure 2 and Figure 5 (in A.4.2). The results for other studies for $n = 100, \theta = 10$ and $n = 50, \theta = 1$ are reported in Table 5 and Table 6 in Appendix A.4.2. We note that our CBMA method provides sets (here, prediction sets are in the form of intervals) for which average length is smaller than all other prediction sets. Also, note that, most of Bayes prediction sets overcover and hence have larger sizes. Conformal Bayes as well as our CBMA prediction sets have valid target coverage. But, on average, CBMA prediction sets have shorter length than CB prediction sets. These results demonstrates the empirical efficiency of CBMA prediction sets over CB prediction sets.

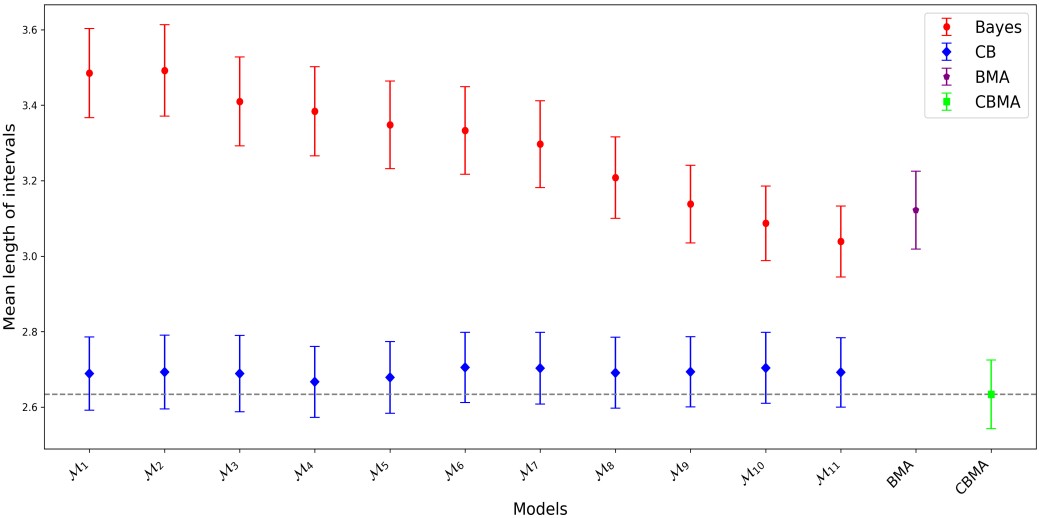

Figure 2: Approximation using Hermite polynomials: Comparison of mean length for prediction sets obtained with different methods: CBMA (proposed method), BMA, individual Bayes prediction sets (in red), and individual conformal Bayes (CB) prediction sets (in blue). Here, we report results based on $E = 50$ number of experiments. We have set the target coverage as $(1 - \alpha) = 0.80$, sample size $n = 100$ and $\theta = 1$.

## 5.3 REAL DATA EXAMPLE: CALIFORNIA HOUSING DATA

We demonstrate our method using the California Housing dataset. The dataset is available in sklearn, and consists of $n = 20640$ subjects, where the continuous response variable denotes the median house value for California districts, expressed in hundreds of thousands of dollars $(100, 000)$, and $d = 8$ covariates, including median income and median house age in block group, average number of rooms per household, block group population. This dataset was derived from the 1990 U.S. census. We first draw random samples of sizes $n = 50, 100$ or $150$ from this dataset for our study. We then standardize all covariates and the response to have mean 0 and standard deviation 1. We consider four different models for these experiments by considering only two covariates in the model. Thus, the models we consider are

$$\mathcal{M}_k: \ y = \beta_0 + \beta_1 X_k + \beta_1 X_{k+1} + \epsilon \, , \ k = 1, 2, 3, 4.$$

The priors we consider on the parameters are $\beta_i \sim N(0, 5)$, $\epsilon \sim N_+(0, 1)$, half-normal distribution. Among all four models, $\mathcal{M}_1$ seems more appropriate as it contains important covariates median income and median house age in block group, as opposed to other covariates. We fit these four models to obtain MCMC samples from the posterior distribution and compute marginal likelihoods for each model. To check coverage, we repeatedly divide our dataset into a training and test dataset for 50 repeats, with $40\%$ of the dataset in the test split. We adopt the setup for simulations similar to Fong & Holmes (2021) in their section 4.1, so that we can compare the performance of our

CBMA method with individual conformal Bayes. We set the miscoverage level at $\alpha = 0.2$. To illustrate the time complexity, we note that, for $n = 150$, the average times for constructing conformal Bayes prediction sets were $0.806(0.004), 0.793(0.002), 0.786(0.002)$ and $0.794(0.003)$ for four models, Bayes credible sets took average times of $0.447(0.003), 0.444(0.002), 0.441(0.002)$ and $0.449(0.004)$. Further, to construct Bayes credible sets using BMA, average additional time was $0.091(0.005)$ and our CBMA method, additional average time was $0.128(0.007)$. We report average lengths and coverages of various prediction sets (prediction intervals in this case) in Figure 3 and 6. We note that, as suggested earlier, model $\mathcal{M}_1$ fits data well as compared to other three models, we expect to have conformal Bayes prediction sets based on this model will be the shortest among all four models. As indicated by our Theorem 2 and Remark 3, our CBMA intervals achieves the shortest lengths as the sample size becomes sufficiently large enough. In our results, we can see CBMA achieves shortest lengths even at finite sample size of $n = 150$. All conformal Bayes approaches as well as CBMA achieves, on average, the target coverage of $1 - \alpha = 0.80$. On the other hand, Bayes models over cover in almost all cases.

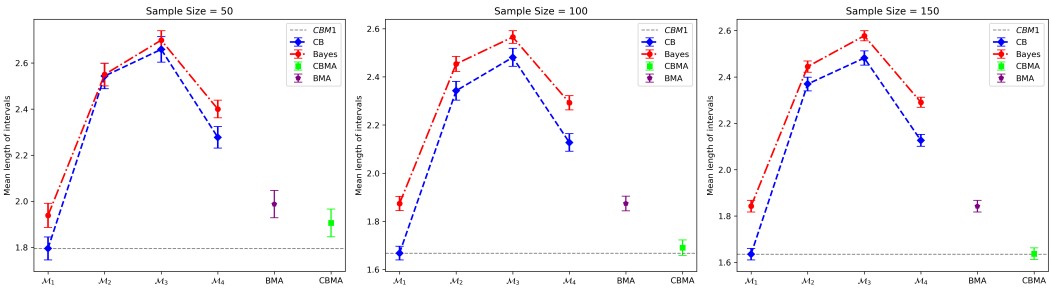

Figure 3: California Housing data: comparison of mean lengths of intervals using Bayes prediction, conformal Bayes (CB) for all four models, CBMA and BMA for different sample sizes.

## 6 CONCLUSIONS

In this work, we identified a challenging issue with the traditional conformal prediction framework: selecting a model in advance in order to construct conformal prediction sets may result in suboptimal prediction sets. In addressing this issue of model uncertainty within the conformal prediction framework, we offer a natural Bayesian solution by integrating Bayesian model averaging in conformal prediction framework. Our proposed CBMA framework seamlessly amalgamates conformity scores from individual Bayesian models to construct a unified conformal prediction set, thus introducing the incorporation of model averaging procedures— a notable advancement currently overlooked from current literature on conformal prediction theory. Additionally, we present an efficient algorithm for constructing CBMA prediction sets, which essentially represent full conformal predictions. Theoretically, we establish the optimal efficiency of our CBMA prediction sets as sample sizes increase, underpinning the efficacy of our approach. Overall, our method holds particular value in real-world scenarios where model uncertainty is a prevalent concern, providing a robust solution within the conformal prediction framework.

However, our approach does have limitations due to the inherent characteristics of MCMC algorithms. The error in these sampling methods can affect prediction sets constructed using CBMA. As noted in Fong & Holmes (2021), we can also expect some robustness to conformal sets as they are based on ranks and not actual values. Furthermore, since we are averaging over multiple models, we expect that the errors in one of the models' training can be masked on averaging.

ACKNOWLEDGMENTS

Bei Jiang and Linglong Kong were partially supported by grants from the Canada CIFAR AI Chairs program, the Alberta Machine Intelligence Institute (AMII), and Natural Sciences and Engineering Council of Canada (NSERC), and Linglong Kong was also partially supported by grants from the

Canada Research Chair program from NSERC. The authors would also like to thank the anonymous reviewers for their constructive comments that improved the quality of this article.

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

## A    APPENDIX

### A.1    PROOF OF LEMMA 1:

(Representation for the conformity measure $\sigma_i^{CBMA}$ in (7) ) Under hierarchical Bayesian model in (6), the posterior predictive density for new text point $(y, x_{n+1})$ can be obtained as follows:

$$
\begin{aligned}
p(Y_i|X_i, Z_{1:n+1}) &= \sum_{k=1}^{K} p(\mathcal{M}_k|Z_{1:n+1})p_{\mathcal{M}_k}(Y_i|X_i, Z_{1:n+1}) \\
&= \sum_{k=1}^{K} \frac{m(Z_{1:n+1}|\mathcal{M}_k)p(\mathcal{M}_k)}{\sum_{k'=1}^{K} m(Z_{1:n+1}|\mathcal{M}_{k'})p(\mathcal{M}_{k'})} p_{\mathcal{M}_k}(Y_i|X_i, Z_{1:n+1}) \\
&= \frac{\sum_{k=1}^{K} m(Z_{1:n+1}|\mathcal{M}_k)p(\mathcal{M}_k)p_{\mathcal{M}_k}(Y_i|X_i, Z_{1:n+1})}{\sum_{k=1}^{K} m(Z_{1:n+1}|\mathcal{M}_k)p(\mathcal{M}_k)} \\
&= \frac{\sum_{k=1}^{K} p(\mathcal{M}_k)\frac{m(Z_{1:n}|\mathcal{M}_k)}{m(Z_{1:n}|\mathcal{M}_k)}m(Z_{1:n+1}|\mathcal{M}_k)p_{\mathcal{M}_k}(Y_i|X_i, Z_{1:n+1})}{\sum_{k=1}^{K} p(\mathcal{M}_k)\frac{m(Z_{1:n}|\mathcal{M}_k)}{m(Z_{1:n}|\mathcal{M}_k)}m(Z_{1:n+1}|\mathcal{M}_k)} \\
&= \frac{\sum_{k=1}^{K} p(\mathcal{M}_k)p_{\mathcal{M}_k}(y|x_{n+1}, Z_{1:n})m(Z_{1:n}|\mathcal{M}_k)p_{\mathcal{M}_k}(Y_i|X_i, Z_{1:n+1})}{\sum_{k=1}^{K} p(\mathcal{M}_k)p_{\mathcal{M}_k}(y|x_{n+1}, Z_{1:n})m(Z_{1:n}|\mathcal{M}_k)} \\
&= \frac{\sum_{k=1}^{K} p(\mathcal{M}_k|Z_{1:n})p_{\mathcal{M}_k}(y|x_{n+1}, Z_{1:n})p_{\mathcal{M}_k}(Y_i|X_i, Z_{1:n+1})}{\sum_{k=1}^{K} p(\mathcal{M}_k|Z_{1:n})p_{\mathcal{M}_k}(y|x_{n+1}, Z_{1:n})} .
\end{aligned}
$$

### A.2    PROOF OF THEOREM 2:

Theorem 3 in Le & Clarke (2022) establishes conditions under which the posterior model probabilities converge to their true values. Below, we summarize key assumptions necessary for proving the theorem, which are also detailed in Le & Clarke (2022).

1. For each $k = 1, \ldots, K$, let $\theta_k \in \Theta_k$, an open subset of $\mathbb{R}^{d_k}$.

2. The densities $p_{\theta_k}(y_i|x_i)$ are measurable in $y$ for all $\theta_k$, $x$ and continuous in $\theta_k$. All second partial derivatives of $p_{\theta_k}(y_i|x_i)$ w.r.t. $\theta_k$ exist and are continuous for all $x, y$, and maybe passed under the integral sign in $\int p_{\theta_k}(y_i|x_i)dy$.

3. There exists a function $K(\cdot)$ such that $E_{\theta_k}K(Y) < \infty$ and each element of the matrix $\left(\frac{\partial^2}{\partial\theta_{k,l}\partial\theta_{k,j}}\log p_{\theta_k}(y_i|x_i)\right)_{l,j=1,\ldots,d_j}$ is absolute value by $K(y)$ uniformly for $\theta_k$ in an open neighbourhood of $\theta_{k,0}$.

4. Assume that, for any $x_i$,

$$
I_i(\theta_{k,0}|x_i) = \left(-E_{\theta_{k,0}}\frac{\partial^2}{\partial\theta_{k,l}\partial\theta_{k,j}}\log p_{\theta_k}(y_i|x_i)\right)_{l,j=1,\ldots,d_k}
$$

is continuous on an open set around the true value $\theta_{k,0}$ and is positive definite at $\theta_{k,0}$.

5. For any $\theta_{k,0} \in \Theta_k$ and any $x_i$, $p_{\theta_k}(y_i|x_i) = p_{\theta_{k,0}}(y_i|x_i)$ a.e. in y implies $\theta_k = \theta_{k,0}$.

6. The average $(1/n) \sum I_i(\theta_{k,0}|x_i)$ is invertible for any $n$ and the $d_k-$ vector $\Psi(\theta_{k,0}|x,y) = \left( \nabla_{\theta_k} \log p_{\theta_{k,0}}(y_i|x_i) \right)^T$ satisfies

$$\sum E \| \frac{1}{n} \left( \sum I_i(\theta_{k,0}|x_i) \right)^{-1/2} \Psi(\theta_{k,0}|x,Y) \|^3 = o\left( \frac{1}{n^{3/2}} \right).$$

7. Priors $\pi_k(\theta_k)$ are continuous for $\theta_k \in \Theta_k$.

Under these conditions, the posterior model probabilities satisfy

$$p(\mathcal{M}_k|Z_{1:n+1}) \to 1 \text{ or } 0 \text{ in probability as } n \to \infty,$$

when $\mathcal{M}_k$ corresponds to the true model $\mathcal{M}_{\text{true}}$ or an incorrect model, respectively. This result directly implies

$$\sigma_i^{CBMA} \to \sigma_i^{\mathcal{M}_{\text{true}}}, \quad \text{in probability, as } n \to \infty.$$

Consequently, the conformal Bayesian model averaging (CBMA) prediction set indicator satisfies

$$\mathbf{1}(y \in C_\alpha^{CBMA}(X_{n+1})) \to \mathbf{1}(y \in C_\alpha^{\mathcal{M}_{\text{true}}}(X_{n+1})) \text{ in probability as } n \to \infty.$$

Thus, the CBMA algorithm (Algorithm 1) produces prediction sets that asymptotically converge to the conformal Bayes prediction set $C_\alpha^{\mathcal{M}\text{true}}(X_{n+1})$ based on the true model $\mathcal{M}_{\text{true}}$.

## A.3 EXPERIMENTS

## A.4 COMPLEXITY OF CONSTRUCTING CBMA INTERVALS

Given the posterior samples and posterior model probabilities, obtaining individual models' conformity scores has a complexity of $\mathcal{O}(n_{\text{grid}} \times \text{posterior samples} \times n_{\text{train}})$ (Fong & Holmes, 2021). The aggregation step (requires dot product of $K$ weights and $K$ conformity scores) has a linear overhead of $\mathcal{O}(K)$, where $K$ is the number of models.

### A.4.1 QUADRATIC MODEL

We now examine the experimental results for the quadratic model, as discussed in Section 5. Figure 4 presents the posterior model probabilities (PMP) for all three models under consideration across $E = 50$ repetitions with a sample size of $n = 100$. The results indicate that Model 1 frequently attains the highest posterior model probability. However, Model 2 also exhibits a substantial number of instances where it achieves a higher PMP. This variability suggests that pre-selecting either Model 1 or Model 2 for constructing conformal prediction sets may lead to suboptimal uncertainty quantification. In contrast, our CBMA method leverages predictions from multiple models, ensuring a more comprehensive integration of model uncertainty. This leads to more efficient and reliable uncertainty quantification. Additionally, Table 2 presents descriptive statistics from these repeated experiments. Notably, CBMA prediction set lengths exhibit a symmetric distribution, whereas CB.M1 prediction sets display a negatively skewed length distribution. This suggests that CBMA results in smaller prediction sets in many instances compared to CB.M1, enhancing efficiency without compromising coverage.

EXECUTION TIMES:

Given the posterior samples obtained from MCMC, we evaluate the computational efficiency of constructing Conformal Bayes (CB) prediction sets and Bayes prediction sets. Table 3 reports the mean execution times (with standard errors in parentheses) for constructing these intervals across different models and sample sizes. Additionally, Table 4 presents the execution times for CBMA and BMA methods. Notably, CBMA incurs higher computational costs compared to BMA, reflecting the additional complexity introduced by model averaging in the conformal framework. However, these times remain negligible relative to the broader MCMC sampling process, making CBMA a computationally feasible approach for uncertainty quantification.

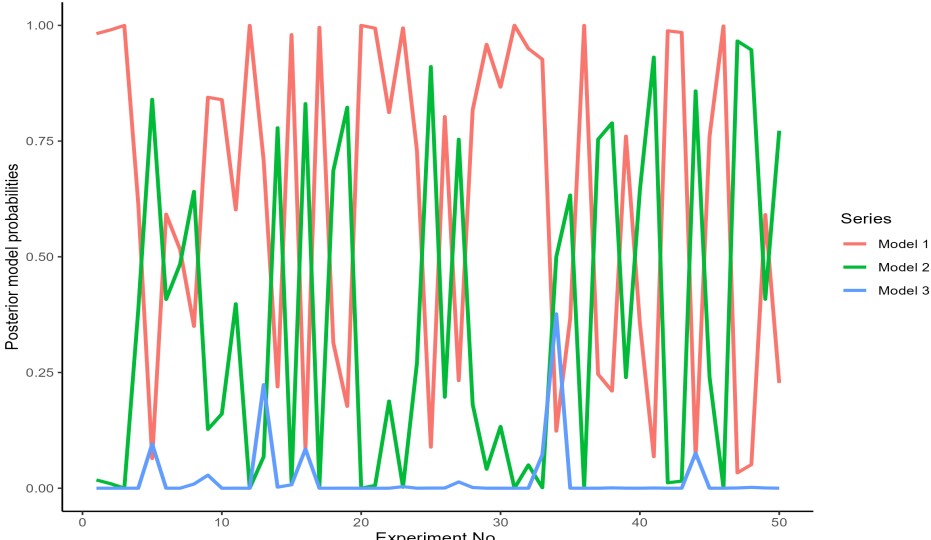

Figure 4: Posterior model probabilities (PMP) for all three models considered in our experiment with quadratic model with sample size $n = 100$.

Table 1: Quadratic model: Comparison of coverage and length for prediction sets obtained with different methods: CBMA (proposed method), BMA, individual Bayes prediction sets BayesM1, BayesM2, BayesM3, individual conformal Bayes prediction sets CBM1, CBM2, CBM3. Here, we report mean and standard error (SE) based on $E = 50$ number of experiments. We have set the target coverage as $(1 - \alpha) = 0.80$.

| Method | CBMA | BMA | BayesM1 | BayesM2 | BayesM3 | CBM1 | CBM2 | CBM3 |
|---|---|---|---|---|---|---|---|---|
| **Coverage** | | | | | | | | |
| $n = 100$ | | | | | | | | |
| Mean | 0.800 | 0.811 | 0.813 | 0.803 | 0.804 | 0.801 | 0.799 | 0.804 |
| SE | 0.010 | 0.010 | 0.010 | 0.010 | 0.009 | 0.010 | 0.011 | 0.010 |
| $n = 200$ | | | | | | | | |
| Mean | 0.807 | 0.811 | 0.813 | 0.813 | 0.805 | 0.812 | 0.815 | 0.805 |
| SE | 0.007 | 0.007 | 0.007 | 0.007 | 0.007 | 0.007 | 0.007 | 0.008 |
| **Length** | | | | | | | | |
| $n = 100$ | | | | | | | | |
| Mean | 0.535 | 0.537 | 0.537 | 0.545 | 0.579 | 0.534 | 0.547 | 0.577 |
| SE | 0.008 | 0.007 | 0.007 | 0.007 | 0.009 | 0.008 | 0.009 | 0.010 |
| $n = 200$ | | | | | | | | |
| Mean | 0.522 | 0.522 | 0.522 | 0.532 | 0.558 | 0.524 | 0.536 | 0.559 |
| SE | 0.006 | 0.004 | 0.004 | 0.005 | 0.006 | 0.006 | 0.007 | 0.007 |

### A.4.2 APPROXIMATION USING HERMITE POLYNOMIALS

The results of studies for $n = 100, \ 50, \ \theta = 10$ and $n = 50, \theta = 1$ are reported in Table 5 and Table 6. We note that CBMA sets have shorter lengths on average in all these experiments.

Table 2: Descriptive statistics for average lengths of prediction sets obtained with different methods: CBMA, BMA, individual Bayes prediction sets Bayes M1, Bayes M2, Bayes M3, individual conformal Bayes prediction sets CB M1, CB M2, CB M3. Here, we report Median, Mean, Standard error (SE), Lower and upper quartiles ($Q_1$ and $Q_3$), Inter-Quantile range (IQR) based on $T = 50$ number of experiments. We have set $\alpha = 0.1$.

| Prediction | CBMA | BMA | Bayes M1 | Bayes M2 | Bayes M3 | CB M1 | CB M2 | CB M3 |
|---|---|---|---|---|---|---|---|---|
| n = 100 | | | | | | | | |
| Median | 0.684 | 0.697 | 0.695 | 0.715 | 0.739 | 0.702 | 0.688 | 0.736 |
| Q1 | 0.64 | 0.651 | 0.652 | 0.666 | 0.706 | 0.643 | 0.64 | 0.666 |
| Q3 | 0.736 | 0.738 | 0.74 | 0.756 | 0.788 | 0.735 | 0.755 | 0.782 |
| IQR | 0.096 | 0.087 | 0.088 | 0.09 | 0.082 | 0.092 | 0.115 | 0.116 |
| Mean | 0.686 | 0.697 | 0.696 | 0.709 | 0.745 | 0.687 | 0.696 | 0.732 |
| SE | 0.01 | 0.008 | 0.008 | 0.009 | 0.008 | 0.009 | 0.01 | 0.011 |
| n = 200 | | | | | | | | |
| Median | 0.676 | 0.676 | 0.675 | 0.69 | 0.706 | 0.671 | 0.681 | 0.709 |
| Q1 | 0.643 | 0.644 | 0.647 | 0.664 | 0.683 | 0.64 | 0.646 | 0.681 |
| Q3 | 0.711 | 0.699 | 0.699 | 0.711 | 0.745 | 0.711 | 0.71 | 0.74 |
| IQR | 0.068 | 0.055 | 0.052 | 0.047 | 0.062 | 0.071 | 0.064 | 0.059 |
| Mean | 0.673 | 0.676 | 0.675 | 0.691 | 0.714 | 0.672 | 0.682 | 0.708 |
| SE | 0.007 | 0.006 | 0.006 | 0.006 | 0.007 | 0.007 | 0.006 | 0.008 |

| Model | n = 100 | n = 200 |
|---|---|---|
| CBM1 | 11.198 (0.071) | 25.470 (0.067) |
| CBM2 | 11.631 (0.075) | 28.040 (0.222) |
| CBM3 | 11.637 (0.140) | 25.950 (0.222) |
| BayesM1 | 4.425 (0.038) | 8.380 (0.040) |
| BayesM2 | 4.641 (0.037) | 9.496 (0.113) |
| BayesM3 | 4.661 (0.061) | 8.956 (0.110) |

Table 3: Mean times (SE) for CB and Bayes prediction intervals for models $(\mathcal{M}_1, \mathcal{M}_1, \mathcal{M}_1)$.

| Model | n = 100 | n = 200 |
|---|---|---|
| CBMA | 0.02397 (0.0014346) | 0.09902 (0.0039647) |
| BMA | 0.00077 (0.0000435) | 0.00123 (0.0000492) |

Table 4: Additional mean times (SE) for CBMA and BMA prediction intervals .

## A.5 REAL DATA EXAMPLE: CALIFORNIA HOUSING DATA

Figure 6 presents the mean coverage probabilities for different methods, with a confidence level of $1 - \alpha = 0.8$. The results indicate that Bayes prediction intervals overcover, providing overly conservative intervals. In contrast, Conformal Bayes (CB) and CBMA consistently achieve the target coverage, ensuring both reliability and efficiency. Additionally, CBMA, by integrating model uncertainty, results in more adaptive and stable prediction intervals compared to individual CB models.

Furthermore, we compare our CBMA set aggregation step with majority vote conformal set aggregation method proposed by Gasparin & Ramdas (2024b). While this method is designed to effectively aggregate sets while preserving the coverage guarantee, it does not necessarily aim to minimize the size of the merged set. In contrast, our CBMA approach offers the additional advantage of allowing the weights in the aggregation step to be learned directly from the same data, addressing a key limitation of existing model aggregation methods. We conducted a direct evaluation by comparing the lengths of CBMA-aggregated intervals to those obtained from the majority vote strategy using individual conformal Bayes prediction sets. For this comparison, we applied Corollary 4.1 from Gasparin & Ramdas (2024b), accounting for the dependency among our individual sets. In our example, the mean ratios of the lengths of intervals from the majority vote procedure relative to

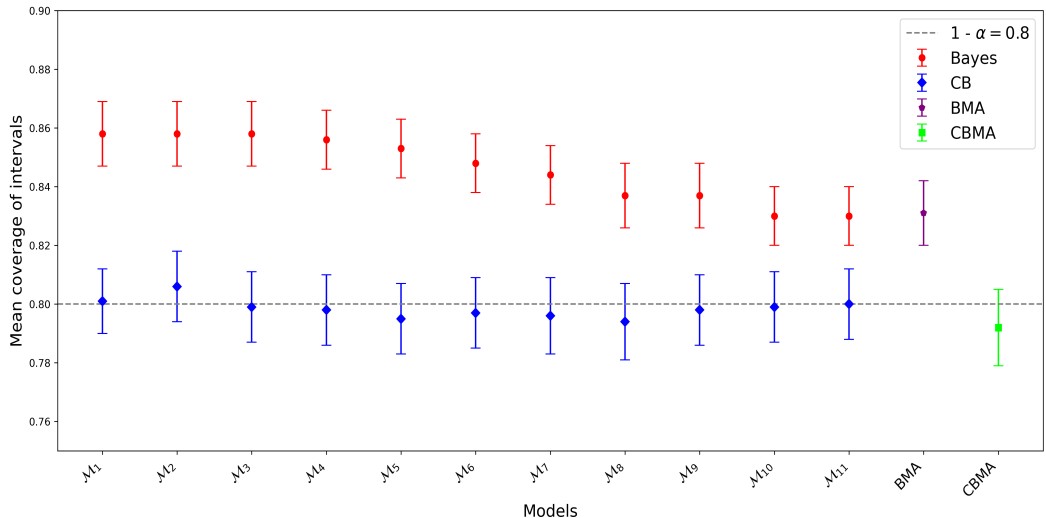

Figure 5: Approximation using Hermite polynomials: Comparison of mean coverage for prediction sets obtained with different methods: CBMA (proposed method), BMA, individual Bayes prediction sets (in red), and individual conformal Bayes (CB) prediction sets (in blue). Here, we report results based on $E = 50$ number of experiments. We have set the target coverage as $(1-\alpha) = 0.80$, sample size $n = 100$ and $\theta = 1$.

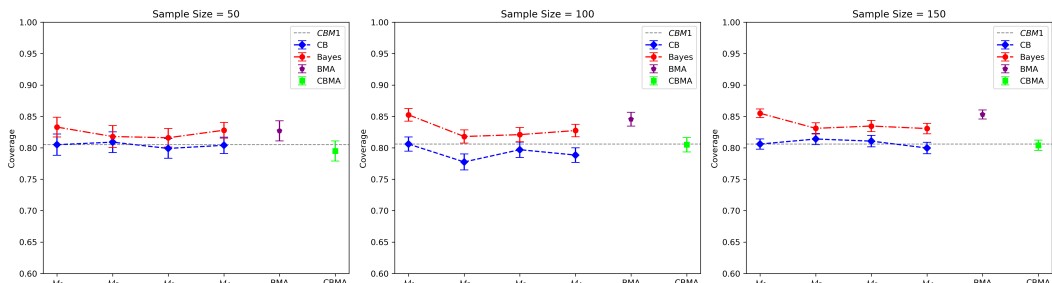

Figure 6: California Housing data: comparison of mean coverages of intervals using Bayes prediction, conformal Bayes (CB) for all four models, CBMA and BMA for different sample sizes. Here, we set $1 - \alpha = 0.8$.

those from CBMA were observed to be $1.0765\,(0.0253)$, $1.1196\,(0.0151)$, and $1.1458\,(0.0133)$ for sample sizes $n = 50$, $n = 100$, and $n = 150$, respectively. These results shows that our CBMA approach gives shorter intervals as compared to majority vote aggregation strategy.

Table 5: Approximation using Hermite polynomials: Comparison of coverage and length for prediction sets obtained with different methods: CBMA (proposed method), BMA, individual Bayes prediction sets (BayesM1-BayesM11), and individual conformal Bayes prediction sets (CBM1-CBM11). Here, we report mean and standard error (SE) based on $E = 50$ number of experiments. We have set the target coverage as $(1 - \alpha) = 0.80$ and $\theta = 1$.

| Method | Length | | | | Coverage | | | |
|--------|--------|--------|--------|--------|----------|--------|--------|--------|
| | n=100 | | n=50 | | n=100 | | n=50 | |
| | Mean | SE | Mean | SE | Mean | SE | Mean | SE |
| BayesM1 | 3.485 | 0.118 | 3.574 | 0.144 | 0.858 | 0.011 | 0.862 | 0.014 |
| BayesM2 | 3.492 | 0.121 | 3.574 | 0.146 | 0.858 | 0.011 | 0.854 | 0.013 |
| BayesM3 | 3.410 | 0.118 | 3.537 | 0.148 | 0.858 | 0.011 | 0.856 | 0.013 |
| BayesM4 | 3.384 | 0.118 | 3.461 | 0.142 | 0.856 | 0.01 | 0.853 | 0.013 |
| BayesM5 | 3.348 | 0.116 | 3.389 | 0.141 | 0.853 | 0.01 | 0.846 | 0.013 |
| BayesM6 | 3.333 | 0.116 | 3.344 | 0.139 | 0.848 | 0.01 | 0.845 | 0.013 |
| BayesM7 | 3.297 | 0.115 | 3.290 | 0.137 | 0.844 | 0.01 | 0.838 | 0.014 |
| BayesM8 | 3.208 | 0.108 | 3.240 | 0.143 | 0.837 | 0.011 | 0.827 | 0.015 |
| BayesM9 | 3.138 | 0.103 | 3.182 | 0.141 | 0.837 | 0.011 | 0.828 | 0.016 |
| BayesM10 | 3.087 | 0.099 | 3.131 | 0.141 | 0.830 | 0.010 | 0.829 | 0.016 |
| BayesM11 | 3.039 | 0.094 | 3.067 | 0.141 | 0.830 | 0.010 | 0.829 | 0.015 |
| CBM1 | 2.689 | 0.097 | 2.864 | 0.128 | 0.801 | 0.011 | 0.819 | 0.017 |
| CBM2 | 2.693 | 0.098 | 2.857 | 0.131 | 0.806 | 0.012 | 0.811 | 0.016 |
| CBM3 | 2.689 | 0.101 | 2.883 | 0.132 | 0.799 | 0.012 | 0.808 | 0.016 |
| CBM4 | 2.667 | 0.094 | 2.877 | 0.137 | 0.798 | 0.012 | 0.819 | 0.015 |
| CBM5 | 2.679 | 0.095 | 2.884 | 0.136 | 0.795 | 0.012 | 0.816 | 0.015 |
| CBM6 | 2.705 | 0.093 | 2.867 | 0.120 | 0.797 | 0.012 | 0.818 | 0.015 |
| CBM7 | 2.703 | 0.095 | 2.873 | 0.120 | 0.796 | 0.013 | 0.818 | 0.015 |
| CBM8 | 2.691 | 0.094 | 2.823 | 0.121 | 0.794 | 0.013 | 0.818 | 0.016 |
| CBM9 | 2.694 | 0.093 | 2.763 | 0.115 | 0.798 | 0.012 | 0.821 | 0.016 |
| CBM10 | 2.704 | 0.094 | 2.740 | 0.112 | 0.799 | 0.012 | 0.822 | 0.015 |
| CBM11 | 2.692 | 0.092 | 2.753 | 0.114 | 0.800 | 0.012 | 0.818 | 0.014 |
| BMA | 3.122 | 0.103 | 3.130 | 0.134 | 0.831 | 0.011 | 0.829 | 0.014 |
| CBMA | 2.634 | 0.091 | 2.683 | 0.118 | 0.792 | 0.013 | 0.808 | 0.015 |

Table 6: Approximation using Hermite polynomials: Comparison of mean Coverage and Length for prediction sets obtained with different methods: CBMA (proposed method), BMA, individual Bayes prediction sets $(BayesM1 - BayesM11)$ , individual conformal Bayes prediction sets $CBM1, \ldots, CBM11$. We have set the target coverage as $(1 - \alpha) = 0.8$. Here, $\theta = 10, K = 11, n = 100$.

| Method | Length | | Coverage | |
|---|---|---|---|---|
| | Mean | SE | Mean | SE |
| B1 | 4.958 | 0.101 | 0.861 | 0.01 |
| B2 | 3.780 | 0.121 | 0.862 | 0.011 |
| B3 | 3.634 | 0.121 | 0.856 | 0.011 |
| B4 | 3.499 | 0.122 | 0.855 | 0.01 |
| B5 | 3.465 | 0.121 | 0.852 | 0.01 |
| B6 | 3.432 | 0.121 | 0.847 | 0.01 |
| B7 | 3.406 | 0.120 | 0.843 | 0.01 |
| B8 | 3.282 | 0.109 | 0.835 | 0.011 |
| B9 | 3.189 | 0.103 | 0.835 | 0.011 |
| B10 | 3.141 | 0.099 | 0.834 | 0.011 |
| B11 | 3.094 | 0.096 | 0.831 | 0.011 |
| CB1 | 3.989 | 0.083 | 0.812 | 0.011 |
| CB2 | 2.922 | 0.091 | 0.805 | 0.010 |
| CB3 | 2.726 | 0.092 | 0.798 | 0.011 |
| CB4 | 2.708 | 0.094 | 0.803 | 0.011 |
| CB5 | 2.707 | 0.091 | 0.799 | 0.011 |
| CB6 | 2.703 | 0.088 | 0.800 | 0.011 |
| CB7 | 2.722 | 0.090 | 0.799 | 0.011 |
| CB8 | 2.722 | 0.090 | 0.800 | 0.011 |
| CB9 | 2.716 | 0.090 | 0.801 | 0.011 |
| CB10 | 2.733 | 0.098 | 0.801 | 0.012 |
| CB11 | 2.718 | 0.098 | 0.800 | 0.012 |
| BMA | 3.163 | 0.099 | 0.834 | 0.010 |
| CBMA | 2.683 | 0.089 | 0.799 | 0.011 |

