# OpenReview forum: "CBMA: Improving Conformal Prediction through Bayesian Model Averaging"
_ICLR.cc/2025/Conference — ICLR 2025 Poster_

### Official Review · Reviewer_bVQG · 2024-10-31

**Soundness:** 2
**Presentation:** 2
**Contribution:** 1
**Rating:** 6
**Confidence:** 3

**Summary:**

The paper leverages Conformal Bayesian and Bayesian Model Averaging to reduce the size of Conformal Prediction sets. The Bayesian posterior of different models is used in a weighted average of Bayesian conformity scores, which produces optimal prediction sets when the average contains the exact model and the sample size is infinite. Unlike previous Conformal Bayesian methods, the proposed scheme allows possible misspecifications in the underlying Bayesian model.

**Strengths:**

- Conformal Bayesian can combine the advantages of frequentist and probabilistic approaches.
- Theoretical results on the optimal efficiency of the prediction sets are rare and challenging to obtain in the CP framework.
- Model averaging and ensemble methods perform well in various setups. Studying CP in that setup may produce powerful uncertainty quantification tools.

**Weaknesses:**

- The technical contribution is to combine existing techniques: the BMA and Conformal Bayes.
- It is unclear whether using the Bayesian conformity score is needed or if the method applies to any conformity score.
- The authors should clarify why averaging the (Bayesian) conformity scores associated with different models is better than computing the conformity score of a Bayesian average of the models.
- The main theoretical result holds in the limit of an infinite sample size. In that limit, Bayesian confidence intervals are also correct. Why would one need CP? Including the exact model in the average may look unrealistic. Does the result generalize to models that are only approximately correct?
- Combining the results of Theorem 3 with Remark 1 to have a direct result on the optimality of the prediction sets would align better with the paper's goals.
- The model has not been tested on real data. For example, the authors could show that the scheme produces smaller intervals than standard CP or any other baseline.

**Questions:**

- Is accessing the model parameters required?
- Is the independence of the noise from X a strict assumption for the method applicability?
- Missing reference number below Eq 1.
- In Section 2.4, you say "We propose to average conformity scores from each model to construct combined conformity score." Is this equivalent to Eq.2? Why don't you condition on X in the equations on page 3?
- What do you mean by 'valid conformity score'?
- How expansive is computing the prediction set?
- Does BMA converge to the true model if $n \to \infty$ because its likelihood increases?
- Is this the first work on Conformal Bayesian that does not assume the underlying Bayesian model is exact?

---

> ### Author Response · Authors · 2024-11-27
>
> Thank you for your constructive feedback and for identifying areas where our manuscript can be improved. Below, we provide point-by-point responses to address the concerns you raised. However, we would like to clarify that many of the points you raised are not inherent weaknesses of our method but rather questions or requests for clarification, which we have addressed. We believe it would be unfair to frame these questions as weaknesses to diminish the value of our contribution.
>
>
> **Weakness 1:** We respectfully disagree with your criticism that the novelty of our work is limited to simply combining existing methods. While our framework does build upon the well-established Bayesian Model Averaging to enhance conformal prediction, this integration represents a significant and novel contribution, as recognized and highlighted by Reviewer 3y9y. Specifically, our work is the **first to provide theoretical guarantees** that ensure the resulting conformal prediction sets are optimally efficient, achieving the shortest expected prediction intervals under model misspecification.
>
> ---
>
>
> **Weakness 2:** Yes, the Bayesian conformity score is needed in our CBMA framework. As we discussed in the paper (lines 184-186), the Bayesian posterior predictive density is the optimal choice for the conformity score. Accordingly, we constructed our framework around this score. This choice enables us to derive the efficiency guarantee in our Theorem $2$, which ensures that the prediction sets produced by our framework are optimal in terms of their expected size (minimized prediction interval length) while maintaining the required coverage probability.
>
> ---
>
> **Weakness 3:** As outlined in Equation (7) of the paper, averaging the Bayesian conformity scores associated with different models is mathematically equivalent to computing the Bayesian average of the conformity scores from individual models. Since these two approaches yield the same result within our framework, we are unsure about the distinction implied in your question. We would appreciate further clarification to address your concern effectively.
>
> ---
>
> **Weakness 4.** While our CBMA framework relies on the standard assumption of infinite sample sizes to establish theoretical guarantees for efficiency and coverage, we emphasize that this is a well-established practice in Bayesian model averaging literature for deriving rigorous theoretical properties. This approach is necessary because Bayesian inference does not inherently align with the frequentist notion of repeated sampling, where guarantees are evaluated under repetitions of the experiment. Instead, Bayesian methods focus on updating beliefs about parameters given observed data. The infinite sample size assumption provides a bridge for deriving frequentist-style guarantees, such as efficiency and coverage, in an asymptotic framework.
>
> However, we demonstrate empirically that CBMA performs effectively even with finite sample sizes. In particular to address your concern, we added Figure $1$ showing the convergence of the mean CBMA prediction set sizes to the optimal prediction set sizes (corresponding to the true model). Convergence is observed with $n=200$, indicating that CBMA achieves efficiency with a finite sample size. Additionally, we included a real data example (California housing data) in the revised Section 5.3. This example shows that CBMA produces shorter prediction intervals than other baseline methods at $n=150$. Notably, Bayesian credible intervals tend to over-cover the data and result in wider, less efficient prediction sets at similar sample sizes.
>
> *Case of approximately correct models:* As demonstrated in another experiment, even when the true model is not included in the model space, CBMA demonstrates robustness against such model specification. In such cases, theoretical analysis suggests that CBMA converges to the model closest to the true model in terms of Kullback-Leibler (KL) divergence. This ensures that **CBMA maintains its coverage guarantee and produces near-optimal prediction sets** if a model exists within the KL divergence neighborhood of the true model. We have added detailed discuss of this observation in the revised version.
>
> In summary, while our theoretical results assume infinite sample sizes, the empirical results confirm that CBMA remains highly effective in finite-sample scenarios, offering both robust coverage and efficiency advantages over Bayesian intervals.
>
> ---
>
> **Weakness 5:** Thank you for the suggestion. There is no Theorem 3 in the paper, but we believe you are referring to Theorem 2. In the revised paper, we have updated theoretical result (Theorem 2) that integrates original Theorem 2 and Remark 1, presenting a unified result for the optimality of prediction sets. As you suggested, this enhancement aligns with the goals of the paper and provides a more comprehensive theoretical foundation.
>
> ---

---

> > ### Author Response · Authors · 2024-11-27
> >
> > **Weakness 6:** Following your suggestion, we have added a real data example (California housing data) in the revised version to demonstrate the practical utility of our method and its performance compared to other baseline methods. Specifically, we compare our CBMA approach with the model aggregation strategy recently proposed by Gasparin and Ramdas (2024b), which combines conformal sets using a majority vote scheme. As expected—since their method focuses on efficient set aggregation rather than achieving efficient prediction sets—our CBMA intervals result in significantly shorter prediction intervals. These results are included in the revised version.
> >
> > Additionally, in the updated Section 2.5, we provide an expanded review of existing model aggregation methods and their limitations. Common limitations of these methods include:
> >
> > 1. **Data splitting requirements or reliance on hold-out calibration sets:** Many baseline methods require splitting the available data, which may not be practical for small sample sizes. In contrast, our CBMA approach utilizes all available data, ensuring no efficiency loss from data splitting.
> >
> > 2. **Lack of theoretical guarantees on efficiency and coverage:** Existing methods often fail to provide rigorous guarantees for both efficiency and statistical coverage. Our CBMA framework, on the other hand, is the first to offer such theoretical guarantees on both optimal efficiency and coverage under potential model misspecification.
> >
> > We believe these additions strengthen the paper and provide a more comprehensive comparison of CBMA with other approaches, further highlighting the advantages of our method.
> >
> > ---
> >
> > **Question 1:** Are you referring to the posterior samples of the model parameters? If so, then yes, our method requires access to the posterior samples of the model parameters.
> >
> > ---
> >
> > **Question 2:** We use the form assuming independence of noise from $X$ to simplify the review of the BMA framework. However, the BMA framework is fully capable of handling scenarios where noise depends on $X$. The independence assumption was introduced only for ease of explanation and does not reflect a limitation of the method itself. Recognizing that this might cause confusion, we have revised this part by general Bayesian model setup in the revised paper (lines 131-133).
> >
> > ---
> >
> > **Question 3:** Thank you for pointing it out. We have added the missing reference below Equation (1).
> >
> > ---
> >
> > **Question 4:** The statement in Section 2.4, ``We propose to average conformity scores from each model to construct a combined conformity score,'' corresponds to Equations (3) and (4), not Equation (2). On Page 3, equations are indeed conditional on $X$ (where $Z_{1:n} = { (X_i, Y_i) }_{i=1}^n$ as in Line 70), although we chose not to explicitly include this notation to maintain simplicity. We acknowledge that this could lead to some confusion, which is why we clarified definitions wherever necessary, e.g., for marginal likelihood on lines 159-160 in the original submission. To avoid any future ambiguity, we have added a statement in the revised paper explicitly addressing this point (line 161 in the revised paper).
> >
> > ---
> >
> > **Question 5:** By a ``valid conformity score,'' we mean a conformity score function that is exchangeable in its first argument. This clarification has been explicitly stated in the revised paper (lines 84-85).
> >
> > ---
> >
> > **Question 6:** Given the posterior samples and posterior model probabilities, computing the conformity scores for individual models involves a complexity of $\mathcal{O}(n_{\text{grid}} \times \text{posterior samples} \times n_{\text{train}})$, as outlined in Fong and Holmes (2021). The model weights can then be efficiently derived using these computed terms. Consequently, the aggregation step, which involves calculating the dot product of $K$ weights and $K$ conformity scores, incurs only a linear computational overhead of $\mathcal{O}(K)$, where $K$ represents the number of models.
> >
> > ---
> >
> > **Question 7:** Bayesian Model Averaging converges to the true model under certain conditions as $n \rightarrow \infty$. This convergence is driven by the combined influence of the likelihood, priors, and the Bayesian updating process, rather than by the likelihood alone. Together, these elements ensure that the posterior probability concentrates on the true model with sufficiently large sample size.
> >
> > ---
> >
> > **Question 8:** Yes, to the best of our knowledge, our work is the first to explore Conformal Bayesian methods without the assumption that the underlying Bayesian model is exactly correct or true. Reviewer $3y9y$ has also acknowledged this contribution.
> >
> > ---
> >
> > **References**
> >
> > Edwin Fong and Chris C Holmes. Conformal bayesian computation. Advances in Neural Information Processing Systems, 34:18268–18279, 2021.
> >
> > Matteo Gasparin and Aaditya Ramdas. Merging uncertainty sets via majority vote. arXiv preprint arXiv:2401.09379, 2024b.
> >
> > ---

---

> > > ### Comment · Reviewer_bVQG · 2024-11-28
> > > **many thanks for the detailed rebuttal**
> > >
> > > I thank the authors for their extensive and clarifying replies and for adding new experiments to the paper. I will rise my score to 6.

---

> > > > ### Author Response · Authors · 2024-11-29
> > > >
> > > > Thank you again for taking the time to evaluate our paper, providing constructive comments that have helped improve our work, and raising the score to 6, which means a lot to us.

---

### Official Review · Reviewer_ctnD · 2024-11-02

**Soundness:** 3
**Presentation:** 2
**Contribution:** 2
**Rating:** 6
**Confidence:** 3

**Summary:**

This paper proposes a solution in the form of Conformal
Bayesian model averaging (CBMA) that combines Bayesian Model Averaging (BMA) with conformal prediction to address the potential issue of suboptimal prediction intervals when the Bayesian model itself is misspecified. The theoretical and empirical results show the efficiency of CBMA.

**Strengths:**

The results presented in the paper confirm the effectiveness of the proposed method.

**Weaknesses:**

1.The novelty of the work is not very significant, as it simply combines several existing methods.

2.The names of the subsections under the "Simulation" section are not entirely appropriate,  since quadratic model is a type of non-linear model. It may be more appropriate to title whether the model is polynomial.

3.The paper seems not explain why the results for model 10 and model 11 are missing when $n=50$ in Table 2.

4.The paper contains a few typographical errors. For example, the last sentence before Section 2 has a typo. Additionally, there is a missing equation reference on line 150.

**Questions:**

1.Is it a clear pattern in the paper for when the authors use the term "prediction set" versus "prediction interval"?

2.Is there a comparison of the time costs between CBMA and other methods?

3.Why are the values of $\alpha$ set as well as the  ratio of the size of training set to test set differently in Table 1 and Table 2? It would be better to unify them or display both values or provide a rationale for why they differ, as this would improve the clarity and comparability of the results.

---

> ### Author Response · Authors · 2024-11-27
>
> Thank you for your constructive feedback and for highlighting areas where our manuscript can be improved. Below, we provide point-by-point responses to address the weaknesses you noted.
>
> **Weakness 1:** We respectfully disagree with your criticism that the novelty of our work is limited to simply combining existing methods. While our framework does build upon the well-established Bayesian Model Averaging to enhance conformal prediction, this integration represents a significant and novel contribution, as recognized and highlighted by Reviewer 3y9y. Specifically, our work is the **first to provide theoretical guarantees** that ensure the resulting conformal prediction sets are optimally efficient, achieving the shortest expected prediction intervals under model misspecification.
>
> ---
>
> **Weakness 2:** Thank you for the suggestion. We have renamed subsection 5.2 to *``Approximation using Hermite Polynomials"* for better clarity.
>
> ---
>
> **Weakness 3:** The number of models used follows the setting $ K = \lfloor 3 n_{\text{train}}^{1/3} \rfloor $, as mentioned in Line 425. For $ n = 50,  n_{\text{train}} = 30 $, resulting in $ K = 9$. These settings align with those in Lu \& Su (2015). But, to avoid the confusion and for the sake of completeness, we have added results for model 10 and model 11 for sample size $n = 50$ in the revised version.
>
> ---
>
> **Weakness 4:** We appreciate the feedback and have corrected the typographical errors, including the missing equation reference and typos before Section 2.
>
> ---
>
> **Question 1:** We have standardized the term *``prediction set"* throughout the paper. While the prediction sets in our experiments are intervals, the theoretical results apply to prediction sets in general.
>
> ---
>
> **Question 2:** We will include execution time details in our simulation experiments to construct various prediction sets (conformal Bayes, Bayes prediction, BMA prediction and our CBMA ) in the Appendix where we provide additional details of our experiments. For the real data example, we will discuss such execution times in the main text to provide times costs between these different approaches.
>
>
> ---
>
> **Question 3:** Following your suggestions, we have added the new experiments in Section 5.1 in the revised paper, with consistent experimental settings with other experiments ($\alpha = 0.2$, train-test split of $60-40\%$), which will improve the clarity and allow comparisons across experiments.
>
> ---
>
> **References:**
>
> Xun Lu and Liangjun Su. Jackknife model averaging for quantile regressions. Journal of Econometrics, 188(1):40–58, 2015.

---

### Official Review · Reviewer_3y9y · 2024-11-03

**Soundness:** 3
**Presentation:** 2
**Contribution:** 2
**Rating:** 6
**Confidence:** 4

**Summary:**

The authors highlight that the efficiency of traditional conformal prediction can degrade in the presence of model uncertainty. To address this, they propose Conformal Bayesian Model Averaging (CBMA), which combines multiple non-conformity scores from a Bayesian perspective. This approach integrates Bayesian model averaging with Bayesian conformal prediction (full conformal Bayes; Fong & Holmes, 2021), allowing it to leverage the optimality of Bayesian model averaging from a theoretical standpoint. From a computational perspective, CBMA utilizes the add-one-in importance sampling approach of Fong & Holmes (2021), enabling it to use all available data for model training while avoiding exhaustive computation. The work demonstrates CBMA’s high efficiency through numerical justification in both correctly specified and misspecified model scenarios.

**Strengths:**

While several works in the field of conformal prediction have applied a Bayesian perspective, this paper is meaningful since it is the first to attempt combining “multiple models” through Bayesian model averaging, marking a novel contribution. The proposed framework is straightforward, integrating traditional Bayesian model averaging directly, yet it is powerful as it incorporates model uncertainty more fully into the framework.

**Weaknesses:**

This work is limited by a lack of discussion comparing its approach to existing model aggregation methods in conformal prediction including frequentist’s perspective. Although Section 2.5 touches on related works, the primary advantage highlighted is the ability to use the full data, which may be better understood as a contribution from Fong & Holmes (2021) rather than an original novelty here. To justify the method empirically, it would be more reasonable to benchmark a few of these existing methods, comparing their coverage, interval length, and execution time with suggested method.

**Questions:**

1. This question relates to the previously mentioned points. What do you consider the distinguishing features of your approach compared to existing conformal set aggregation methods? Were any experiments conducted to compare them? If so, what were the results, and if not, what are your thoughts on this?

  2. In the simulations, the nominal value of $\alpha$ was set to 0.1 in the linear model scenario, but 0.2 in the nonlinear case. Is there a particular reason for these distinct choices? Also, did you observe any systematic performance changes in the model based on the value of $\alpha$?

---

> ### Author Response · Authors · 2024-11-27
>
> Thank you for acknowledging the contribution of our work. Below we provide answers to address your specific concerns.
>
> **Weaknesses:** Thank you for your thoughtful feedback regarding comparisons with existing frequentist approaches for model aggregation in conformal prediction. Based on your suggestions, we have revised our paper to include an **expanded literature review of these frequentist approaches** and a detailed discussion of the distinct contributions and scope of our Bayesian Model Averaging (BMA) framework within the conformal prediction literature.
>
> We emphasize that **frequentist methods** often focus on effective set aggregation or selection but **lack theoretical guarantees** of optimal efficiency and valid statistical coverage. Our CBMA framework, on the other hand, directly addresses these gaps by providing theoretical guarantees for both optimal efficiency and valid statistical coverage in the resulting prediction sets under potential model misspecification.
>
> For example, in the context of transductive conformal methods (also known as full conformal methods, which make full use of the data for prediction),  Yang and Kuchibhotla (2024) provide selection rules to choose the smallest valid conformal prediction sets. However, their methods may suffer from coverage loss and lack theoretical guarantees for achieving statistical efficiency in the selected prediction set. In the context of inductive conformal methods (also known as split conformal methods, which require splitting the available data into training and calibration sets, reducing efficiency), prior works such as Vovk (2015), Linusson et al. (2017), Carlsson et al. (2014), Toccaceli and Gammerman (2019) propose methods that combine p-values generated by different models or data splits to create aggregated predictions. However, these combined p-values may not be well-calibrated, leading to prediction sets that do not accurately represent the intended coverage level. This lack of calibration undermines the reliability of the aggregated predictions.
>
> Given these fundamental differences, direct comparisons between CBMA and frequentist model aggregation methods are not meaningful, as they lack rigorous guarantees of efficiency and coverage. For instance, we compared the lengths of CBMA aggregated intervals with those of conformal sets aggregated using the majority vote strategy (applying Corollary 4.1 from Gasparin and Ramdas (2024b) due to dependency among individual sets) in our real data example. The observed mean ratios of interval lengths constructed using majority vote scheme and our CBMA approach for $n = 50, 100, 150$ are $1.0765 (0.0253)$, $1.1196 (0.0151)$, and $1.1458 (0.0133)$, respectively, indicating that CBMA leads too shorter intervals. These results have been included in the revised paper's real data example section.
>
> ---
>
> **Question 1:** We have revised Section 2.5 to provide a **more detailed comparison** of our method with existing approaches, highlighting their contributions, limitations, and how CBMA addresses these gaps. This **expanded discussion** clarifies the unique contributions of our work relative to the existing literature. Additionally, to address your concern about the weakness of our paper above, we have elaborated on why existing frequentist methods are not directly comparable to our approach, given the fundamental differences in objectives and guarantees.
>
> ---
>
> **Question 2:** Following your suggestions, we have added the new experiments in Section 5.1 in the revised paper, with **consistent experimental settings** with other experiments ($\alpha = 0.2$, train-test split of $60-40$), which will improve the clarity and allow comparisons across experiments.
>
> ---
>
> **References:**
>
> Lars Carlsson, Martin Eklund, and Ulf Norinder. Aggregated conformal prediction. In Artificial Intelligence Applications and Innovations: AIAI 2014 Workshops: CoPA, MHDW, IIVC, and MT4BD, Rhodes, Greece, September 19-21, 2014. Proceedings 10, pp. 231–240. Springer, 2014.
>
> Matteo Gasparin and Aaditya Ramdas. Conformal online model aggregation. arXiv preprint
> arXiv:2403.15527, 2024a.
>
>
> Matteo Gasparin and Aaditya Ramdas. Merging uncertainty sets via majority vote. arXiv preprint arXiv:2401.09379, 2024b.
>
> Henrik Linusson, Ulf Norinder, Henrik Bostr¨om, Ulf Johansson, and Tuve L¨ofstr¨om. On the cali-
> bration of aggregated conformal predictors. In Proceedings of the Sixth Workshop on Conformal and Probabilistic Prediction and Applications, volume 60 of Proceedings of Machine Learning Research, pp. 154–173. PMLR, 13–16 Jun 2017.
>
> Vladimir Vovk. Cross-conformal predictors. Annals of Mathematics and Artificial Intelligence, 74: 9–28, 2015.
>
> Yachong Yang and Arun Kumar Kuchibhotla. Selection and aggregation of conformal prediction sets. Journal of the American Statistical Association, pp. 1–13, 2024. doi: 10.1080/01621459.
> 2024.2344700.

---

> > ### Comment · Reviewer_3y9y · 2024-11-27
> >
> > Thank you for conducting the additional numerical experiments as requested! These have helped address some of the uncertainties. However, I still remain hesitant about the claim that other methods with the same objectives lack theoretical guarantees. In particular, Liang et al. (2024; https://arxiv.org/abs/2408.07066) specifically address the coverage loss highlighted in Yang & Kuchibhotla (2024). As a result, I am still unconvinced regarding the reliance on multiple models and the computational demands of methods such as MCMC, or other Bayesian computations.

---

> > > ### Author Response · Authors · 2024-11-27
> > >
> > > Thank you for bringing up the additional reference. However, we must clarify that the work by Liang et al. (2024) is dealing with a completely different goal than ours. Liang et al. (2024) aims to **``constructing a finite sample valid conformal prediction set while selecting a model that minimizes the size of the prediction set, from a given family of pre-trained models”**. In other words, their efficiency guarantee relies on knowing the true model.
> > >
> > > In contrast, our work aims to provide optimally efficient conformal prediction sets without requiring knowledge of the true model. Our efficiency guarantee holds with potential model misspecification. Additionally, while methods such as Yang & Kuchibhotla (2024) suffer from selection bias and therefore lack statistical validity for the resulting prediction sets, Liang et al. (2024) seeks to correct such biases to achieve validity. In comparison, our method inherently achieves finite-sample validity through a probabilistic aggregation approach based on Bayesian Model Averaging (BMA), avoiding the need for bias correction entirely.

---

> > > > ### Comment · Reviewer_3y9y · 2024-11-28
> > > >
> > > > Thank you for addressing my concerns. You pointed out the coverage loss problem in Yang & Kuchibhotla’s (2024) work. I would like to highlight that Liang et al. (2024) addressed this issue by restoring valid prediction sets, demonstrating that methods for selection and aggregation across multiple models to construct valid prediction sets have been actively studied.
> > > >
> > > > However, the theoretical properties of CBMA’s efficiency, as well as the broader additional numerical experiments you conducted, effectively showcase the strengths of CBMA. Based on this, I am raising my score from 5 to 6.

---

> > > > > ### Author Response · Authors · 2024-11-28
> > > > > **Thank you**
> > > > >
> > > > > Thank you for your positive remarks on the theoretical efficiency properties of CBMA and the additional numerical experiments we conducted to demonstrate its strengths. Your decision to raise your score from 5 to 6 means a great deal to us. We sincerely thank you for taking the time to evaluate our paper and for providing constructive comments that have helped improve our work.

---

### Official Review · Reviewer_Nttv · 2024-11-03

**Soundness:** 3
**Presentation:** 4
**Contribution:** 2
**Rating:** 6
**Confidence:** 4

**Summary:**

The authors, working in the niche of bayesian conformal prediction, propose a bayesian model averaging procedure "infused" with conformal prediction, in order to recover coverage in the case of model misspecification

**Strengths:**

I found the article quite well written.
Moreover, the proposal is of clear theoretical and applied interest, and developed with care and mathematical soundness.

**Weaknesses:**

I fear that the authors have oversold their claim. In fact, in the literature proposals of conformal model averaging can be found  (https://www.sciencedirect.com/science/article/pii/S0925231219316108, https://arxiv.org/abs/2408.06642#:~:text=Unlike%20traditional%20methods%2C%20conformal%20ensembling,%2Dto%2Dinterpret%20uncertainty%20estimates. to name a few).

I found the simulation quite lacking in terms of depth, as well and presentation of the results

**Questions:**

-As shown in the "weaknessess", some proposals (usually in a frequentist setting) are present in the literature when it comes to model averaging/ensembling in a conformal sense. Can you comment on this, maybe providing a more thorough analysis of the outstanding literature?

- I find the cases provided by the authors of limited applicative interest, I wonder if it could be possible to provide more challenging "misspecification" cases, like for instance cases where the model misspecification causes the conditional iid assumption to break (e.g. an AR2 model with only one lag).

- The impact in terms of interval length does not seem statistically significant. Are the authors able to check this, using maybe the Scheffe' method?

---

> ### Author Response · Authors · 2024-11-27
>
> Thank you for your constructive feedback. Below, we provide point-by-point responses to address the weaknesses you noted.
>
> **Weakness 1:**  You are correct that conformal prediction based on model averaging has gained attention, as the papers that you provided demonstrate how incorporating model averaging can help reduce the volume of prediction sets through experiments. However, none of these prior works provide a theoretical guarantee. Our work is **the first to establish such a guarantee**, specifically proving that the resulting conformal prediction intervals converge to the optimal level of efficiency.
> As Reviewer 3y9y noted, our approach is the first to combine ``multiple models'' in conformal framework through Bayesian model averaging, marking a significant and novel contribution. For this reason, we believe our contribution has been accurately stated.
> Thank you for providing the references. We have acknowledged these prior works on conformal model averaging in Section 2.5 and highlighted their limitations in the revised paper.
>
> **Weakness 2:** We appreciate your feedback. Our **simulations** were intentionally designed to **validate the theoretical guarantees** of our method and provide comparisons with other approaches in a simple yet illustrative manner, which allows readers to easily recognize that the candidate models are mis-specified, yet our proposed method consistently outperforms others by producing the shortest prediction intervals while maintaining coverage probabilities close to the nominal level. To further demonstrate the practical utility of our method, we have added a **real data example** in the revised version of the paper and showed that CBMA works effectively. Finally, to improve the presentation of results, we have added **plots** of comparisons for mean lengths and coverages for prediction intervals in our simulation studies.
>
> **Question 1:** Following your suggestion, we have thoroughly reviewed the relevant literature, including your recommended papers, Linusson et al. (2020) and Yang and Kuchibhotla (2024), as well as several others in Section 2.5. We have expanded our discussion to provide a **more comprehensive discussion of existing methods**, highlighting their limitations and how they compare to and differ from our approach. We believe this expanded discussion addresses your concerns and clarifies the unique contributions of our work in relation to the existing literature.
>
> **Question 2:** We appreciate your thoughtful feedback and understand your interest in exploring more challenging ``misspecification'' cases, such as the AR2 model. However, conducting such experiments may not be necessary in this context, as it is well-known in the literature on Bayesian Model Averaging (BMA) for time series data that if all candidate models fail to account for temporal dependence, BMA will lead to biased estimates and underestimated uncertainty. In other words, efficiency gains are only achievable when the misspecified models still capture temporal dependence to some extent. However,  models with temporal dependence structure violate the exchangeability condition required by our method.
> To address your concern that our method may have limited applicative interest, we have **added new experiment** to evaluate the performance of our approach on real dataset, where the true model is unknown. This represents a fair assessment of our method's effectiveness, as it mirrors real-world scenarios where all methods may potentially mis-specify the true model. These new results are included in the revised paper.
>
> **Question 3:** We appreciate your suggestion, as it offers an insightful way to evaluate whether the prediction intervals produced by our method are likely the shortest among all methods, when not knowing the population truths. Specifically, Scheffé's method tests whether the expected lengths of prediction intervals (i.e., the population truths) differ across methods.
> However, we have already demonstrated in Remark 1 of the paper that the expected interval lengths are theoretically guaranteed to be the shortest under our method. Since the population truth is already known in this context, there is no need to rely on hypothesis testing, which is inherently subject to type I and type II errors and could lead to incorrect conclusions about the population. In conclusion, our **theoretical guarantee makes additional hypothesis testing unnecessary.**

---

> > ### Comment · Reviewer_Nttv · 2024-11-27
> >
> > I thank the authors for the kind reply, but while I appreciate the effort, I remain unconvinced on some issues, more specifically Weakness 1 and Question 3
> >
> > With respect to Question 1, as shown by other reviewers, some methods do have theoretical guarantees, so I believe a more proper contextualisation of contributions should be in order.
> >
> > WRT to Question 3, your theoretical guarantees on minimal size are asymptotic, and nothing is stated on the rate of convergence nor on finite sample properties... I was thus proposing tests to assess, in finite samples, what is the behaviour of your aggregation procedure and, e.g., what is the minimum sample size after which your proposal starts to yield statistically significant results.

---

> ### Author Response · Authors · 2024-11-28
> **Thank you for your follow up questions.**
>
> **Revisit Question 1**
>
> Thank you for your feedback. As noted in the manuscript and our responses, we conducted a thorough review of relevant works and highlighted either their distinct goals or their lack of theoretical guarantees compared to ours. We also appreciate Reviewer 3y9y pointing out another recent unpublished paper by Liang et al. (2024; https://arxiv.org/abs/2408.07066). However, as we clarified to Reviewer 3y9y, Liang et al. (2024) addresses a completely different goal. Their work focuses on constructing a finite-sample valid conformal prediction set while selecting a model that minimizes the size of the prediction set from a given family of pre-trained models. In other words, their efficiency guarantee relies on knowing the true model.
>
> In contrast, our work provides optimally efficient conformal prediction sets without requiring knowledge of the true model. Our efficiency guarantee holds even under potential model misspecification. This critical distinction was acknowledged by Reviewer 3y9y, who raised their score from 5 to 6, recognizing the unique strengths and contributions of our CBMA framework.
>
> **Revisit Question 3**
>
> As noted in our responses, we appreciate your suggestion on using hypothesis testing to determine whether the prediction sets obtained using our CBMA framework is the shortest with finite sample size. However, such an approach does suffer from type I and type II error, like all hypothesis tests do. We do acknowledge that our theoretical guarantees on minimal size are asymptotic; in our original submission, we already empirically assessed the finite-sample performance of our CBMA framework in the original submission. These results demonstrate that CBMA performs effectively in finite samples, producing shorter prediction intervals than baseline methods while maintaining valid coverage.
>
> That said, we have taken your suggestion seriously and expanded our empirical evaluation in the revised paper to include both additional simulations and a newly added real-world example. Instead of focusing on hypothesis testing, we have added figures illustrating the convergence of the expected sizes of CBMA prediction sets to the optimal prediction set sizes (corresponding to the true model). In summary, convergence is observed at $n=200$ in simulations and $n=150$ in the real-world example, demonstrating that CBMA achieves optimal efficiency with finite sample sizes.
>
> Lastly, while your request regarding the rate of convergence is beyond the primary scope of our work, we acknowledge its importance. To address this concern, we note that Equation (7) suggests that the rate of convergence corresponds to the convergence rate of posterior model probabilities, a well-studied topic in Bayesian literature (see, for instance, Rossell, 2022).
>
> We hope this additional clarification and empirical evidence address your concerns.
>
>
> **Reference:**
>
> Rossell, D. (2022). Concentration of posterior model probabilities and normalized $L_0$ criteria. Bayesian Analysis, 17(2), 565-591.

---

> > ### Comment · Reviewer_Nttv · 2024-11-28
> >
> > I thank the authors for the response, that are now satisfactory.
> > I have thus raised my evaluation from 5 to 6.
> > Good luck with the final selection!

---

> > > ### Author Response · Authors · 2024-11-28
> > >
> > > We sincerely appreciate your thoughtful feedback and are glad that our responses addressed your concerns satisfactorily. We are also very grateful for your decision to raise your evaluation score from 5 to 6. Thank you for your best wishes, and we hope for a positive outcome in the final selection process!

---

### Meta-Review · Area_Chair_WWrA · 2024-12-24

**Metareview:**

The formal novelties of the paper lie in its integration of Bayesian Model Averaging  with conformal prediction to address model misspecification and improve prediction set efficiency. By aggregating conformity scores from multiple Bayesian models using posterior model probabilities, the proposed Conformal Bayesian Model Averaging ensures valid frequentist coverage while optimizing the expected volume of prediction sets. The method theoretically guarantees convergence to optimal prediction sets as sample size increases, when the true model is in the model space (which apparently holds even under potential model misspecification).

It is to be noted that Theorem 2 (consequence of  Le & Clarke (2022) is not distribution-free). I also did not understand the claim when the model is not well specified. The Bayesian Posterior Convergence is that the Posterior probabilities of the true model (if it exists in the candidate set) converge to 1 as the sample size (ok then the quantiles estimates converges to the one of the ground-truth and the convergence of the size). If the true model is not present, posterior probabilities concentrate on the candidate model closest to the true model in terms of (KL) divergence. In the later case, The size of the CBMA prediction set may be larger than the optimal size that would be achieved if the true model were included. Equation 9 in the proof is not established in this case (BTW, the overall proof should be improved).

> In contrast, our work provides optimally efficient conformal prediction sets without requiring knowledge of the true model. Our efficiency guarantee holds even under potential model misspecification.

I could not see any proof of such statement in the paper. When the model is not well specified, the consistency results does not hold and this claim is misleading. Furthermore, what the authors want to claim seems quite straightforward: *any consistent estimation of the ground-truth distribution would produce asymptotically efficient uncertainty set*. This has fairly much nothing to do with model averaging but only consistency.

Along with lack of comparisons to existing methods, the proposed benchmark (e.g. the main Table 1) are quite unconvincing: all the methods achieve very similar size (one needs to look at 3rd digit to notice any difference between the lengths and this is even not significant given the standard deviation).

----
*After further discussions with the SAC, we suggest an Accept, leaving the final decision to the program chairs to potentially bump down. All reviewers raised their scores to 6 ("Marginally above the acceptance threshold") after acknowledging the authors' efforts to address concerns and improve the manuscript. While some issues remain, the paper has merits and could provide valuable insights to the community.*

**Additional Comments On Reviewer Discussion:**

The discussions with reviewers highlighted both the strengths and limitations of the proposed CBMA framework. Reviewers appreciated the theoretical rigor and the novel integration of Bayesian Model Averaging (BMA) with conformal prediction, providing guarantees of efficiency and valid coverage even under model misspecification. However, concerns were raised about the incremental nature of the contribution, limited empirical evaluation, and lack of direct comparisons with existing frequentist and Bayesian methods. Specific issues included insufficiently challenging misspecification scenarios, unclear advantages over simpler aggregation strategies, and reliance on strong assumptions, such as the presence of the true model in the candidate set. The authors addressed these by expanding the empirical evaluations, including real-world datasets, improving clarity in theoretical arguments, and elaborating on distinctions from prior work. While these efforts satisfied some reviewers, leading to score increases, others maintained reservations about the method’s practical applicability and incremental contributions.

> Set $\alpha=0.1$ or smaller as it is done in common in the literature.

---

### Decision · Program_Chairs · 2025-01-22

Accept (Poster)